*Report*

EMBO
Molecular Medicine

# Structure-guided design of a PfCyRPA-based vaccine against blood-stage malaria

Nawsad Alam [iD] [1,2], Clare Wolfle [iD] [1,2], Egle Butkeviciute[1,2], Doris Quinkert[1,2], Lloyd D W King[1,2] & Matthew K Higgins [iD] [1,2 ✉]

## Abstract

**Effective vaccines against malaria are urgently required. All components of the PfPCRCR complex are essential for erythrocyte invasion by *Plasmodium falciparum* and are potential vaccine immunogens against blood-stage malaria. Of these, PfRH5 has progressed furthest in clinical development, while PfCyRPA also induces parasite growth-inhibitory antibodies. Here, we used direct nanoparticle coupling and structure-guided design to generate improved PfCyRPA-based immunogens. PfCyRPA is a six-bladed β-propeller. Blades 1 and 2 are exposed in the PfPCRCR complex and contain the epitopes of the most potent known growth-inhibitory antibodies. We therefore performed structure-guided design to generate a correctly folded, thermostable epitope mimic, PfCyRPA-EM, containing blades 1 and 2. In a pre-clinical model, PfCyRPA-EM elicited antibodies that inhibited parasite growth at lower concentrations than those elicited by PfCyRPA. In addition, the higher thermostability of PfCyRPA-EM and its improved expression as an I53-50 nanoparticle fusion make it well-suited for clinical development, alone or with other immunogens.**

**Keywords** Malaria; Erythrocyte-invasion; PfCyRPA; Structure-guided Vaccine Design
**Subject Categories** Microbiology, Virology & Host Pathogen Interaction; Structural Biology

## Introduction

Malaria due to infection with *Plasmodium falciparum* continues to cause a huge global health burden, with an estimated 597,000 deaths and 263 million cases reported in 2023 (World Health Organisation, 2024). The WHO recently recommended two malaria vaccines, Mosquirix™ (RTS,S/AS01) (RTS,S Clinical Trials Partnership, 2015) and R21/Matrix-M® (Datoo et al, 2024), which target the pre-erythrocytic stage of the parasite life cycle. However, we urgently need vaccines with shorter dosing regimens, higher efficacy, and longer-lasting protection. The blood stage of the parasite is a promising target for a vaccine to prevent both disease and transmission, and an effective blood-stage vaccine could be used alone or combined with an existing pre-erythrocytic vaccine for increased efficacy (Duffy et al, 2024).

The PfPCRCR complex (Farrell et al, 2024; Scally et al, 2022) is critical for erythrocyte invasion, and its components, PfRH5, PfCyRPA, PfRIPR, PfCSS, and PfPTRAMP, are promising candidates for blood-stage vaccine development. Of these, PfRH5 has progressed the furthest. Pre-clinical studies showed that full-length PfRH5 (known as the RH5.1 immunogen) can induce growth-inhibitory antibody responses in rodents (Douglas et al, 2011; Wright et al, 2014), Aotus (Douglas et al, 2019; Douglas et al, 2015) and humans (Alanine et al, 2019; Barrett et al, 2024; Natama et al, 2024; Wang et al, 2024).

Immunisation with RH5.1, formulated with Freund's adjuvant, protected Aotus from challenge with *Plasmodium falciparum* (Douglas et al, 2019; Douglas et al, 2015). Although this vaccine did not confer protective immunity against parasite challenge in human adult volunteers from a non-malaria endemic setting, it slowed parasite growth, providing the first demonstration of human in vivo activity for a blood-stage malaria vaccine (Minassian et al, 2021). However, immunisation of 5–17-month-old children with RH5.1/Matrix-M® in a phase 1b study in a malaria-endemic setting elicited higher growth-inhibitory antibody responses, reaching the level of growth-inhibitory activity (GIA) associated with the protection of Aotus from challenge with *Plasmodium falciparum* (Silk et al, 2023). Moreover, a recent phase 2b clinical trial in a seasonal malaria setting with RH5.1/Matrix-M® reported an efficacy of 55% against clinical malaria in 5–17-month-old children compared with the control group when administered with a delayed third dose and 40% when using a monthly regimen (Natama et al, 2024). In parallel, structure-guided vaccine design has been employed to generate improved vaccine immunogens, with a truncated, thermally stabilised version of PfRH5 (RH5.2) (Campeotto et al, 2017) giving greater quantitative antibody immunogenicity in pre-clinical models (King et al, 2024) and with a single epitope-based vaccine eliciting a high-quality antibody response (Harrison et al, 2024). With next-generation PfRH5-based vaccines entering clinical trials, PfRH5 remains the most advanced candidate for a blood-stage malaria vaccine.

[1]Department of Biochemistry, Dorothy Crowfoot Hodgkin Building, University of Oxford, South Parks Road, Oxford OX1 3QU, UK. [2]Kavli Institute for Nanoscience Discovery, Dorothy Crowfoot Hodgkin Building, University of Oxford, South Parks Road, Oxford OX1 3QU, UK. ✉E-mail: matthew.higgins@bioch.ox.ac.uk

While development and testing of PfRH5-based vaccines continue, a second promising approach is to combine the best PfRH5-based immunogen with additional immunogens based on other components of PfPCRCR. Indeed, monoclonal antibodies targeting PfRH5 and PfCyRPA can act synergistically to prevent parasite growth (Alanine et al, 2019; Williams et al, 2024), and a combination of PfRH5 with a fusion protein containing PfCyRPA and a fragment of PfRIPR improved the level of in vitro parasite growth inhibition compared to RH5.1 alone in a pre-clinical model (Williams et al, 2024). However, more complex multicomponent vaccines come with potential for immune interference, as well as increasing complexity and cost of production and distribution. It is therefore imperative for each component to be designed to induce a focused immune response and to show increased stability and simplified production.

Structural studies of PfCyRPA revealed it to adopt a six-bladed β-propeller architecture (Chen et al, 2017; Favuzza et al, 2017; Ragotte et al, 2022). When bound to PfRH5 and PfRIPR, the surfaces of blades 1 and 2 of PfCyRPA are exposed (Farrell et al, 2024). These same surfaces contain epitopes for all parasite growth-inhibitory monoclonal antibodies characterised to date, while non-inhibitory antibodies bind other surfaces of PfCyRPA (Chen et al, 2017; Favuzza et al, 2017; Ragotte et al, 2022). The inhibitory antibodies prevent erythrocyte binding by PfRCR and prevent parasite growth through a steric mechanism (Farrell et al, 2024). PfCyRPA-binding antibodies can function synergistically, with lateral antibody-antibody contacts for antibodies bound to neighbouring epitopes, which stabilise the immune complex and increase the apparent antibody affinity (Ragotte et al, 2022).

These findings suggest that the best PfCyRPA-based immunogen will contain just blades 1 and 2 of PfCyRPA. Display of this immunogen on a virus-like nanoparticle scaffold is expected to specifically elicit growth-inhibitory antibodies without eliciting non-inhibitory antibodies that target blades 3–6 of PfCyRPA. In addition, such an immunogen will contain a sufficiently large antigenic surface to elicit antibodies that synergise through heterotypic interactions. A recent study tested a peptide consisting of an unaltered form of blades 1 and 2 of PfCyRPA, but this was poorly immunogenic (Bjornsson et al, 2024). Indeed, such an immunogen would be unlikely to fold correctly and would not correctly present conformational epitopes for growth-inhibitory antibodies when removed from the context of the PfCyRPA β-propeller. We therefore used structure-guided design, coupled with directed evolution, to generate a correctly folded version of PfCyRPA blades 1 and 2. Here we present the design, characterisation, and pre-clinical testing of this immunogen, showing it to induce a high-quality growth-inhibitory antibody response.

## Results

### PfCyRPA is more immunogenic when fused directly to I53-50 nanoparticles than when conjugated to HBsAg via SpyTag-SpyCatcher

Before starting the structure-guided design of improved PfCyRPA-based immunogens, we assessed the effect of conjugation of PfCyRPA to two distinct nanoparticle platforms. Multivalent antigen presentation on nanoparticles often results in substantially enhanced immunogenicity,

most likely due to more effective B-cell receptor clustering (Bachmann and Jennings, 2010; Bachmann et al, 1993; Nguyen and Tolia, 2021). A wide range of platforms have been successfully tested, including both naturally occurring and rationally designed nanoparticles (Nguyen et al, 2021). In addition, different approaches have been used to conjugate antigens to these particles, including expression as direct antigen-nanoparticle fusion (Marcandalli et al, 2019; Walls et al, 2020) or by conjugating tagged antigens to preformed nanoparticles, such as through the SpyTag-SpyCatcher system (Brune et al, 2016). In this study, we selected a version of PfCyRPA (residues D29–E362), with potential N-linked glycosylation sites removed through mutations S147A, T342A and T340A as, like other *Plasmodium falciparum* surface proteins, PfCyRPA is not naturally glycosylated when endogenously expressed. We conjugated this to different nanoparticle display systems and assessed which elicited the most growth-inhibitory antibody response.

We selected two different systems. We firstly selected the hepatitis B surface antigen (HBsAg) nanoparticle displaying SpyCatcher (King et al, 2024). This system allows SpyTagged antigens to be conjugated to preformed nanoparticles, and we previously used this nanoparticle to display PfRH5-based immunogens (Harrison et al, 2024). We were able to conjugate SpyTag-fused PfCyRPA to these nanoparticles with ~85% efficiency to form PfCyRPA-HBsAg (Table EV1; Fig. EV1A,B). We also tested the computationally designed protein nanoparticle I53-50 (Walls et al, 2020). This contains two components, I53-50A.1NT1 and I53-50B.4PT1, which rapidly assemble in vitro into nanoparticles which allow the conjugation of 60 antigens per nanoparticle. PfCyRPA was fused to I53-50A.1NT1 via a flexible glycine-serine linker and was expressed in Expi293F cells to generate PfCyRPA-I53-50A. I53-50B.4PT1 was purified from *E. coli* and assembled in vitro with purified PfCyRPA-I53-50A by equimolar mixing to form PfCyRPA-I53-50, resulting in nanoparticles with a PfCyRPA conjugation efficiency of 100% (Table EV1; Fig. EV1A,B).

We next used dynamic light scattering (DLS) to assess the degree of homogeneity of each PfCyRPA-conjugated nanoparticle. PfCyRPA-I53-50 displayed a narrow distribution of hydrodynamic radius, indicating a homogeneous particle population (Fig. EV1C). In contrast, PfCyRPA-HBsAg exhibited a broader size distribution, as measured by DLS (Fig. EV1C), most likely due to the inherent size variability of unconjugated HBsAg nanoparticles. Surface plasmon resonance analysis (SPR) showed that both nanoparticles bound to growth-inhibitory PfCyRPA targeting antibodies Cy.003, Cy.004, and Cy.007, indicating exposure of all three epitopes in the conjugated nanoparticles (Fig. EV1D).

We next studied the ability of PfCyRPA-I53-50 and PfCyRPA-HBsAg to induce parasite growth-inhibitory antibodies. We used a standard model in which rats are immunised with vaccine immunogens and purified IgG is tested using a growth-inhibition assay, which measures the reduction in *Plasmodium falciparum* parasite growth in human red blood cells in the presence of antibodies. This model has been shown to be predictive of the efficacy of vaccines in immunised Aotus and humans (Douglas et al, 2015; Natama et al, 2024; Silk et al, 2024). In the absence of a readily scalable animal model for blood-stage malaria, this is the standard pre-clinical screening tool.

We immunised groups of six rats with three doses (days 0, 28, 56) of either PfCyRPA-I53-50 or PfCyRPA-HBsAg, both formulated with the human-compatible Matrix-M® adjuvant with matched antigen dose (Fig. 1A). We measured the quantity of

PfCyRPA-specific IgG elicited after the final dose by ELISA, showing that PfCyRPA-I53-50 induced an almost twofold higher titre of PfCyRPA-specific antibodies than PfCyRPA-HBsAg (Fig. 1B). We also assessed the growth-inhibitory activity of these antibodies. Total IgG was purified from rat sera from day 70 and GIA was performed at a twofold serial dilution of antibody concentrations starting at 6 mg/ml (Fig. 1C). The antibody concentration required for 50% growth-inhibitory activity ($EC_{50}$) was approximately twofold lower in sera from PfCyRPA-I53-50-immunised rats compared to those immunised with PfCyRPA-HBsAg (Fig. 1D). The enhanced immunogenicity of PfCyRPA on the I53-50 platform, along with the simplicity of its production, led us to continue further immunogenicity studies using this platform.

## Structure-guided design of a focused immunogen based on PfCyRPA

We next aimed to use insights from structures of PfCyRPA in complex with growth-inhibitory antibodies to design an improved PfCyRPA-based vaccine immunogen. The epitopes for the structurally characterised growth-inhibitory antibodies that bind PfCyRPA are predominantly contained within blades 1 and 2 of the six-bladed structure of PfCyRPA, with the Cy.004 and Cy.007 epitopes fully recapitulated in blades 1 and 2 (Fig. 2A). We therefore reasoned that an immunogen comprising solely blades 1 and 2 would elicit a more focused growth-inhibitory antibody response than full-length PfCyRPA.

We began by testing the expression of a polypeptide containing blades 1 and 2 of PfCyRPA (residues I42–S152, incorporating the S147A mutation, Table EV2). This construct was not secreted by Expi293F cells, most likely due to poor folding. Indeed, removal of blades 1 and 2 from their context within the PfCyRPA β-propeller will expose hydrophobic surfaces which would normally contact blades 3 and 6. To generate an improved version of blades 1–2, we first used the PROSS protein repair server (Campeotto et al, 2017; Goldenzweig et al, 2016), aiming to introduce changes that enhance its stability and expression levels while preventing residues which form the epitope surfaces from changing (Fig. 2B; Table EV2). Although the PROSS design demonstrated significant improvements in expression in Expi293F cells, it had two limitations that prevented it from meeting our design specifications. First, it formed a dimer (Fig. EV2A; Table EV3). Second, its binding affinities for both Cy.004 and Cy.007 were ~15-fold and ~90-fold lower than those for full-length PfCyRPA (Fig. EV2B; Table EV4).

To improve on the PROSS design, we conducted directed evolution, combining error-prone PCR and yeast surface display library screening. A gene library was generated by random error-prone PCR of the PROSS design and was transfected into EBY100 yeast cells. We used antibody binding to select yeast cells from the library, sorting first by Magnetic-activated cell sorting (MACS) using Cy.004 and then by seven rounds of fluorescence-activated cell sorting (FACS) with Cy.004 and Cy.007. Cy.004 was used in early rounds as it has the higher growth-inhibitory activity (Farrell et al, 2024; Ragotte et al, 2022), with Cy.007 used in subsequent rounds to ensure selection of immunogens with both epitope surfaces recapitulated (Fig. EV3). This produced a cell population that bound to both antibodies with significantly higher affinities than the PROSS design displayed on yeast cells (Fig. 2C).

After the final sorting round, three unique sequences, BDP-YD1-1, BDP-YD1-2 and PfCyRPA-EM, representing different sequence clusters, were selected for expression and biophysical assessments (Table EV2). The designs BDP-YD1-1 and PfCyRPA-EM showed good expression in Expi293F cells (Fig. EV2C), and their binding affinities against Cy.004 and Cy.007 were measured using SPR (Figs. 2D and EV2B). This led to the selection of PfCyRPA-EM (Fig. 2B; Table EV2), which bound both antibodies with similar affinities as PfCyRPA (Fig. 2D; Table EV4). While the affinities for Cy.004 were very similar (1.86 nM vs 2.19 nM), indicating preservation of the epitope, Cy.007 bound to PfCyRPA-EM with a three-fold lower $K_D$ than to PfCyRPA (4.60 nM vs 1.44 nM), due to a faster off-rate. Circular dichroism (CD) analysis revealed PfCyRPA-EM to have a ~ 14 °C higher thermal stability than PfCyRPA (Fig. 2E). PfCyRPA-EM exhibited an excellent expression profile in *E. coli* (Fig. EV2C), unlike PfCyRPA, which did not express in a soluble form. We also crystallised PfCyRPA-EM bound to the Fab fragment of Cy.004 and determined its structure by molecular replacement, revealing PfCyRPA-EM to closely resemble PfCyRPA blades 1 and 2, with a root mean square deviation of 0.879 Å (Fig. 2F; Table EV5). Indeed, the epitopes for both Cy.004 and Cy.007 were not significantly different in PfCyRPA and PfCyRPA-EM, suggesting that the small change in off-rate for Cy.007 might be due to increased flexibility in Cy.007 epitope, rather than misfolding. Therefore, PfCyRPA-EM is a soluble, stable, readily expressed protein which adopts the same structure as blades 1 and 2 of PfCyRPA.

## PfCyRPA-EM induces a higher quality immune response than PfCyRPA

We next compared the outcome of immunising rats with either PfCyRPA-EM or PfCyRPA, each fused to I53-50. PfCyRPA-EM was fused to I53-50A.1NT1 via a flexible glycine-serine linker, and the fusion was expressed in Expi293F cells at levels approximately sixfold higher than PfCyRPA-I53-50A.1NT1 (Table EV2; Fig. EV4A). In addition, PfCyRPA-EM fused to I53-50A.1NT1 could be expressed in *E. coli*, unlike PfCyRPA-I53-50A.1NT1 (Fig. EV4A). This improved expression in multiple systems means that PfCyRPA-EM-based nanoparticles will be more cost-effective for production than those containing PfCyRPA. PfCyRPA-EM fused to I53-50A.1NT1 was assembled with I53-50B.4PT1 to form PfCyRPA-EM-I53-50 nanoparticle (Fig. EV4B). These were highly homogeneous, as measured by DLS (Fig. EV4C) and bound to growth-inhibitory antibodies Cy.004 and Cy.007, as measured by SPR (Fig. EV4D), demonstrating correct epitope exposure.

Groups of six rats were immunised with either three doses (days 0, 28, 56) of PfCyRPA-EM-I53-50 or three doses of PfCyRPA-I53-50, both formulated with Matrix-M® adjuvant (Fig. 3A). ELISA was then used to quantify the PfCyRPA and PfCyRPA-EM specific IgG elicited by each immunogen. In each case, the anti-PfCyRPA response increased after the first and second, but not third dose, indicating that a two-dose regimen might be sufficient to test the functional actively of the maximal immune response (Fig. 3B). In rats immunised with PfCyRPA-EM, the titre of antibodies against PfCyRPA-EM increased after a third dose, without increasing the PfCyRPA titre, suggesting that the third dose primarily increased levels of antibodies that target regions other than the epitopes of interest (Fig. 3C). Comparison of the titres against PfCyRPA

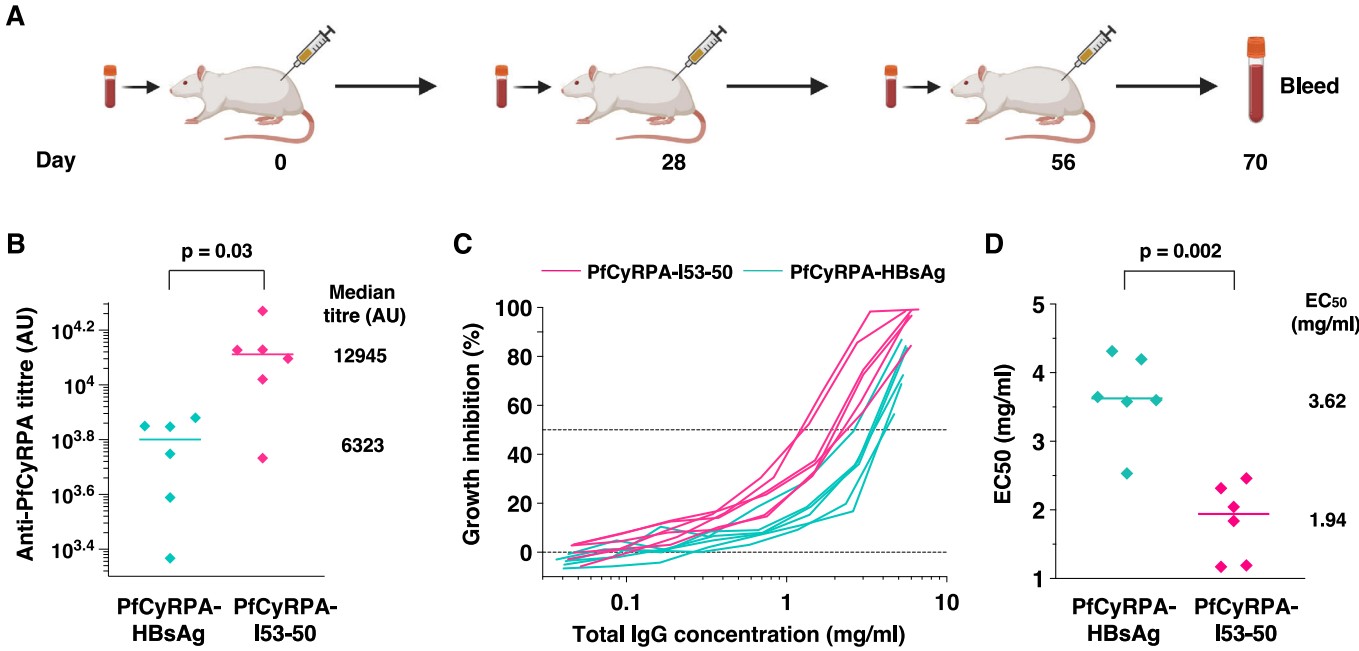

**Figure 1. Immunogenicity of the PfCyRPA-I53-50 and PfCyRPA-HBsAg nanoparticles.**

(A) Rat immunisation schedule with PfCyRPA-I53-50 and PfCyRPA-HBsAg nanoparticles formulated with 25 µg of Matrix-M®. Immunisations were conducted on days 0, 28, and 56, and a final antibody sample was taken on day 70. (B) Antibodies from day 70 sera from rats immunised with PfCyRPA-I53-50 (pink) and PfCyRPA-HBsAg (turquoise) were assessed for their binding to immobilised PfCyRPA by ELISA, with each point representing an individual animal. (C) Growth-inhibitory activity of a twofold dilution series starting at 6 mg/ml of total IgG purified from sera raised by immunisation of rats with PfCyRPA-I53-50 (pink) and PfCyRPA-HBsAg (turquoise), with each line representing an individual animal. Each point is an average of three measurements. (D) Calculated $EC_{50}$ values from (C). In each group, we immunised six rats, and each point represents the data from a single rat. Each point is an average of three technical replicates. Statistical significance was determined using a two-tailed Mann–Whitney test. $P$ values are indicated on the plots. Source data are available online for this figure.

showed that PfCyRPA-EM-I53-50 induced a mean anti-PfCyRPA response ~6.3-fold lower than PfCyRPA-I53-50.

We next evaluated the quality of the induced antibodies using a growth inhibition assay. Our initial focus was on the growth-inhibitory activity of total IgG, as this is most representative of sera and most relevant to the protective response. Total IgG was purified from rat sera on day 70, and GIA was performed using a twofold serial dilution series of these antibodies starting at 6 mg/ml (Fig. 3DE). There was no statistically significant difference between the mean $EC_{50}$ values for the PfCyRPA-I53-50 group and the PfCyRPA-EM-I53-50 group ($P = 0.0649$). This contrasts with a study in which a version of blades 1 and 2 of CyRPA, which was not reengineered to adopt the correct fold, was compared with CyRPA. In this case, the $EC_{50}$ for growth inhibition of IgG induced by the blades 1 and 2 construct was ten-fold greater than that of IgG induced by PfCyRPA (Bjornsson et al, 2024), highlighting the value of rational design to obtain correctly folded immunogens.

In addition to assessing total IgG, we evaluated growth-inhibitory activity after calibrating for the concentration of PfCyRPA-specific antibodies (Williams et al, 2024) rather than total IgG. This allows us to compare the quality of the PfCyRPA antibodies to assess whether the focused PfCyRPA-EM immunogen specifically elicits growth-inhibitory antibodies. ELISA was used to measure the concentration of PfCyRPA-specific antibodies in both sera, allowing us to plot GIA against the titre of PfCyRPA-specific antibodies (Fig. 3F,G). In this analysis, PfCyRPA-specific antibodies from PfCyRPA-EM-I53-50-immunised rats exhibited an $EC_{50}$ around 6.9-fold lower

($P$ value = 0.0087) than those from PfCyRPA-I53-50-immunised rats (Fig. 3G). Therefore, a larger fraction of PfCyRPA-targeting antibodies induced by PfCyRPA-EM-I53-50 are growth inhibitory, resulting in a similar serum antibody functional activity elicited by a lower PfCyRPA-specific antibody titre.

# Discussion

An effective blood-stage malaria vaccine must elicit both a high quantity and high quality of parasite growth-inhibitory antibodies, which can act during the brief time window while the parasite is exposed between egress from one erythrocyte and invasion of another. To improve the quantity of an antibody response, display of antigens on nanoparticles has become increasingly popular as this can increase immunogenicity compared to use of monomeric antigens. To improve antibody quality, structure-based design can be used to generate focused immunogens that present only epitopes of the most protective antibodies, thereby eliciting the most protective antibodies while avoiding ineffective antibodies, or those which might interfere with neutralising responses. In this study, we explored both strategies for PfCyRPA.

We first evaluated two nanoparticle display platforms for their ability to meet our design specifications. An effective display system will induce the most growth-inhibitory antibody response, but this may need to be balanced against the ease of manufacture and distribution. A system in which antigens and nanoparticles are

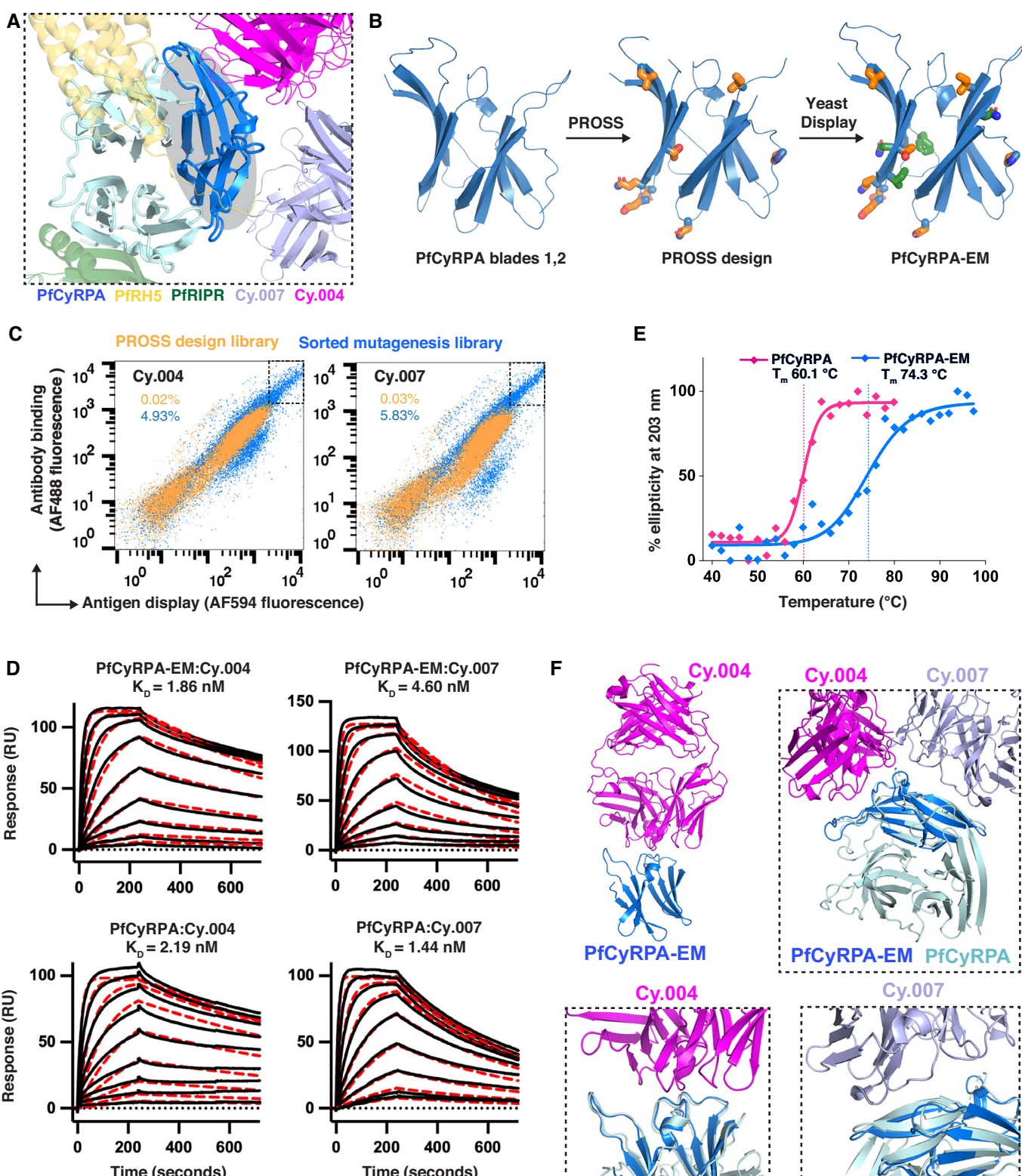

produced separately and then conjugated together after production may generate an effective immune response, but it will be more complex to produce, with possible batch-to-batch variability in the level of conjugation than a direct fusion-based system. Here, we tested both types of systems for PfCyRPA and direct fusion to I53-50 was superior for all criteria, generating more PfCyRPA-specific antibodies which inhibited growth at approximately twofold lower concentration and with simpler manufacture.

**Figure 2. Design, biophysical, and structural characterisation of the PfCyRPA epitope mimic.**

(**A**) A composite model of PfCyRPA bound to PfRH5 (yellow), PfRIPR (green), and the Fab fragments of growth-inhibitory monoclonal antibodies Cy.004 (pink) and Cy.007 (pale blue) based on overlay of experimental structures. CyRPA is shown in pale blue with blades 1 and 2 coloured in darker blue and highlighted with a grey oval. This is the region selected for epitope mimic design. (**B**) Scheme for the immunogen design. A construct containing blades 1 and 2 was redesigned using PROSS to improve expression, followed by yeast surface display to optimise oligomeric state and antibody binding affinity, generating PfCyRPA-EM. (**C**) FACS traces showed improvement in antibody binding following mutagenesis and sorting. The trace for PROSS design (orange) is compared with the outcome of the final sort (blue) with sorting for binding to Cy.004 (left) and Cy.007 (right). Cells with high antibody binding and high surface expression are highlighted with dashed rectangles (top right), with the corresponding percentage indicated on the left side of each panel. (**D**) Surface plasmon resonance analysis of PfCyRPA-EM (top) or PfCyRPA (bottom) binding to Cy.004 (left) and Cy.007 (right), with the data shown as black lines and fits to a 1-to-1 binding model shown as red dashed lines (representative from $n = 2$). (**E**) Circular dichroism melting curves for PfCyRPA (pink) and PfCyRPA-EM (blue), showing increased thermal stability (representative from $n = 2$). (**F**) The crystal structure of PfCyRPA-EM (blue) bound to the Fab fragment of growth-inhibitory monoclonal antibody Cy.004 (pink) is shown in the top left panel. The top right panel shows a composite model of PfCyRPA-EM (blue) aligned to that of PfCyRPA (pale blue), both bound to Cy.004 (pink) and Cy.007 (violet). The lower panels show the same composite model focused on Cy.004 (bottom left) and Cy.007 (bottom right), showing the degree of similarity of PfCyRPA and PfCyRPA-EM in the epitope regions. Source data are available online for this figure.

We next used structure-guided design to generate a synthetic variant of PfCyRPA which contains only blades 1 and 2 of the β-propeller. While simply removing this region of PfCyRPA from its context results in an incorrectly folded protein, the combination of the PROSS protein repair server and selection for growth-inhibitory antibody binding using yeast display and directed evolution generated an immunogen which we call PfCyRPA-EM. Structural studies and antibody binding experiments demonstrated that this folds correctly. When compared with PfCyRPA, PfCyRPA-EM showed an ~14 °C increase in thermal stability, approximately sixfold improvement in expression in mammalian cells and could be expressed as an I53-50 fusion in *E. coli*. When assessed in our pre-clinical model, IgG induced by PfCyRPA-EM-I53-50 immunisation were more specific to PfCyRPA but less abundant than those induced by PfCyRPA-I53-50, resulting in a similar parasite growth-inhibition potency. Therefore, PfCyRPA-EM is an improvement on PfCyRPA in both the quality of antibody response which it induces and in its ease of production and distribution as a vaccine.

The components of the PfPCRCR complex are among the most promising candidates for blood-stage malaria vaccines. While the PfRH5-based immunogen RH5.1 has now shown efficacy in children in a seasonal malaria setting, protection is incomplete, durability is uncertain, and it did not provide protective immunity in malaria challenge studies in adults. While we continue to use structure-guided design to generate improved versions of PfRH5, a combination with other components of PfPCRCR is also a promising strategy. PfCyRPA-EM, with its high stability, ease of production as a two-component nanoparticle and ability to induce a high-quality antibody response, is now available to test in combination with future PfPCRCR-based vaccines. This has the potential to significantly enhance vaccine quality, leading to improved protection against malaria.

## Methods

### Reagents and tools table

| Reagent/resource | Reference or source | Identifier or catalogue number |
|---|---|---|
| **Experimental models** | | |
| *ClearColi* BL21 (DE3) | Cambridge Bioscience | 60810-1 |
| Expi293F™ cells | ThermoFisher | A14527 |
| FreeStyle™ 293-F cells | ThermoFisher | R79007 |
| Female Wistar IGS rats | Noble Life Sciences | N/A |
| *Plasmodium falciparum* 3D7 strain | https://www.beiresources.org/Catalog/BEIParasiticProtozoa/MRA-102.aspx | N/A |
| **Recombinant DNA** | | |
| pEt15b vector | Novagen | 69661 |
| pHLsec vector | Addgene | 99845 |
| **Antibodies** | | |
| Cy.003 | Ragotte et al, 2022 | N/A |
| Cy.004 | Ragotte et al, 2022 | N/A |
| Cy.007 | Ragotte et al, 2022 | N/A |
| Mouse anti-c-Myc 9E10 | Bio-Rad | MCA2200 |
| AlexaFluor 594-conjugated secondary anti-mouse IgG antibody | Thermo Fisher Scientific | A-11032 |
| AlexaFluor 488-conjugated secondary anti-human IgG | Thermo Fisher Scientific | A11013 |
| Goat anti-rat IgG secondary antibody | Sigma | A8438 |
| **Chemicals, enzymes, and other reagents** | | |
| Ni-NTA agarose | Qiagen | 30210 |
| HiTrap™ Protein G HP column | Cytiva | GE29-0485-81 |
| Immobilised Papain | ThermoFisher | 20341 |
| HiTrap™ rProtein A column | Cytiva | GE17-0403-03 |
| Superdex 200 Increase 10/300 GL column | Cytiva | GE28-9909-44 |
| Superdex 75 Increase 10/300 GL column | Cytiva | GE29148721 |
| Superose 6 Increase 10/300 GL column | Cytiva | GE29091596 |
| Recombinant Protein A/G | Cytiva | 21186 |

| Reagent/resource | Reference or source | Identifier or catalogue number |
|---|---|---|
| Matrix-M adjuvant | Novavax | N/A |
| MagnaBind™ Protein G Beads | ThermoFisher | 21349 |
| GeneMorph II Random Mutagenesis Kit | Agilent | 200550 |
| **Software** | | |
| PROSS | Goldenzweig et al, 2016 | N/A |
| JalView | https://www.jalview.org | N/A |
| DIALS v3.0 | Winter et al, 2022 | N/A |
| AIMLESS v0.73 | Evans and Murshudov, 2013 | N/A |
| PHASER MR v2.8.3 | McCoy et al, 2007 | N/A |
| COOT v 0.8.9.2 | Emsley et al, 2010 | N/A |
| Phenix v1.21 | Liebschner et al, 2019 | N/A |
| **Other** | | |
| CM5 series S sensor | Cytiva | BR100530 |

## Protein expression and purification

A synthetic gene encoding PfCyRPA (residues D29–E362), with predicted N-linked glycosylation sites removed by mutations S147A, T340A, and T342A, was fused at the C-terminus to a SpyTag followed by a C-tag. This construct was cloned into the pHLsec vector, which includes an N-terminal secretion signal, and transiently expressed in Expi293F cells using the Expi293 Expression System Kit (Thermo Fisher Scientific). Culture supernatant was collected seven days post-transfection, filtered through a 0.45-µm membrane, and incubated with CaptureSelect C-tagXL affinity resin (Thermo Fisher Scientific), which had been pre-equilibrated with phosphate-buffered saline (PBS), pH 7.4. After washing the resin with 10 column volumes of PBS, bound protein was eluted with 10 column volumes of C-tag elution buffer (20 mM Tris, pH 7.4, 2 M $MgCl_2$). The eluate was further purified by size-exclusion chromatography using a Superdex 200 Increase 10/300 column equilibrated in PBS. SpyCatcher-HBsAg was expressed and purified as described previously (Marini et al, 2019).

To produce I53-50A.1NT1 fused to PfCyRPA or PfCyRPA-EM, constructs were generated by fusing the respective antigens to the N-terminus of I53-50A.1NT1 via a $(GSG)_5G$ linker, followed by a C-terminal hexa-histidine tag for affinity purification. The genes encoding these fusion proteins were cloned into the pHLsec vector containing an N-terminal secretion signal and were transiently expressed in Expi293F cells using the same protocol as for SpyTagged PfCyRPA expression. Culture supernatants were collected seven days post-transfection, filtered through a 0.45 µm membrane, and incubated with HisTrap Excel resin (Cytiva) pre-equilibrated with PBS. After washing the resin with 10 column volumes of PBS containing 20 mM imidazole, bound proteins were eluted using PBS supplemented with 500 mM imidazole before size-

exclusion chromatography using a Superdex 200 Increase 10/300 column equilibrated in PBS.

The gene encoding I53-50B.4PT1 with a C-terminal C-tag was cloned into the pET15b vector and transformed into *ClearColi* BL21 (DE3) competent cells (ClearColi™ Expression Technology). Protein expression was induced at an optical density at 600 nm ($OD_{600}$) of 0.6 by the addition of IPTG to a final concentration of 0.5 mM, and cells were harvested after 3.5 h of incubation at 37 °C. Cell pellets were resuspended in lysis buffer (25 mM Tris, pH 7.4, 150 mM NaCl, 0.75% CHAPS) and lysed by sonication. The clarified lysate was incubated with CaptureSelect C-tagXL affinity resin (Thermo Fisher Scientific). After binding, the resin was washed with 10 column volumes of lysis buffer, and the bound protein was eluted using 10 column volumes of C-tag elution buffer (20 mM Tris, pH 7.4, 2 M $MgCl_2$), before size-exclusion chromatography using a HiLoad 16/600 Superdex 200 pg column equilibrated with 25 mM Tris, pH 7.4, 400 mM NaCl, and 0.75% CHAPS.

Endotoxin was removed from all purified proteins using Pierce High-Capacity Endotoxin Removal Spin Columns (Thermo Fisher Scientific).

### PROSS design of PfCyRPA blades 1 and 2

A model of PfCyRPA residues I42–S152, including the S147A substitution to remove a putative N-linked glycosylation site, was extracted from the crystal structure of PfCyRPA (PDB ID: 7PHW) (Ragotte et al, 2022) using PyMOL. The resulting fragment was submitted to the PROSS (Protein Repair One Stop Shop) server (https://pross.weizmann.ac.il/) using default parameters. The designed model, named 'PROSS design', containing eight mutations, all outside the epitope regions, was selected for expression and biophysical assessment.

### Yeast surface display-based directed evolution

To further improve the folding and antibody binding of the PROSS design, a surface-displayed mutant library was generated in *Saccharomyces cerevisiae* strain EBY100. Random mutagenesis of the gene encoding the PROSS design was performed using error-prone PCR (GeneMorph II Random Mutagenesis Kit, Agilent Technologies). The resulting PCR products were co-electroporated with a linearised pETcon vector containing homologous overlapping regions into EBY100 cells, yielding a library of $\sim 2 \times 10^7$ electroporated cells. The library was initially enriched by magnetic-activated cell sorting (MACS) using the antibody Cy.004 bound to Dynabeads Protein G (Thermo Fisher Scientific). This was followed by seven rounds of fluorescence-activated cell sorting (FACS) using a S3e Cell Sorter (Bio-Rad) to select for variants displaying improved cell surface expression and enhanced binding to antibodies Cy.004 and Cy.007. A C-terminal myc tag was included to enable detection of surface expression using a mouse anti-myc antibody (Bio-Rad) and AlexaFluor 594-conjugated anti-mouse secondary antibody (Thermo Fisher Scientific). Binding to antibodies Cy.004 and Cy.007 was detected using an AlexaFluor 488-conjugated secondary anti-human IgG antibody (Thermo Fisher Scientific). From each FACS round, a fraction of the total sorted cells displaying both high cell surface expression and strong binding to Cy.004 or Cy.007 were selected to the next round. In the first round, cells were bound to Cy.004 at a concentration of 1.0 µg/ml, and the top

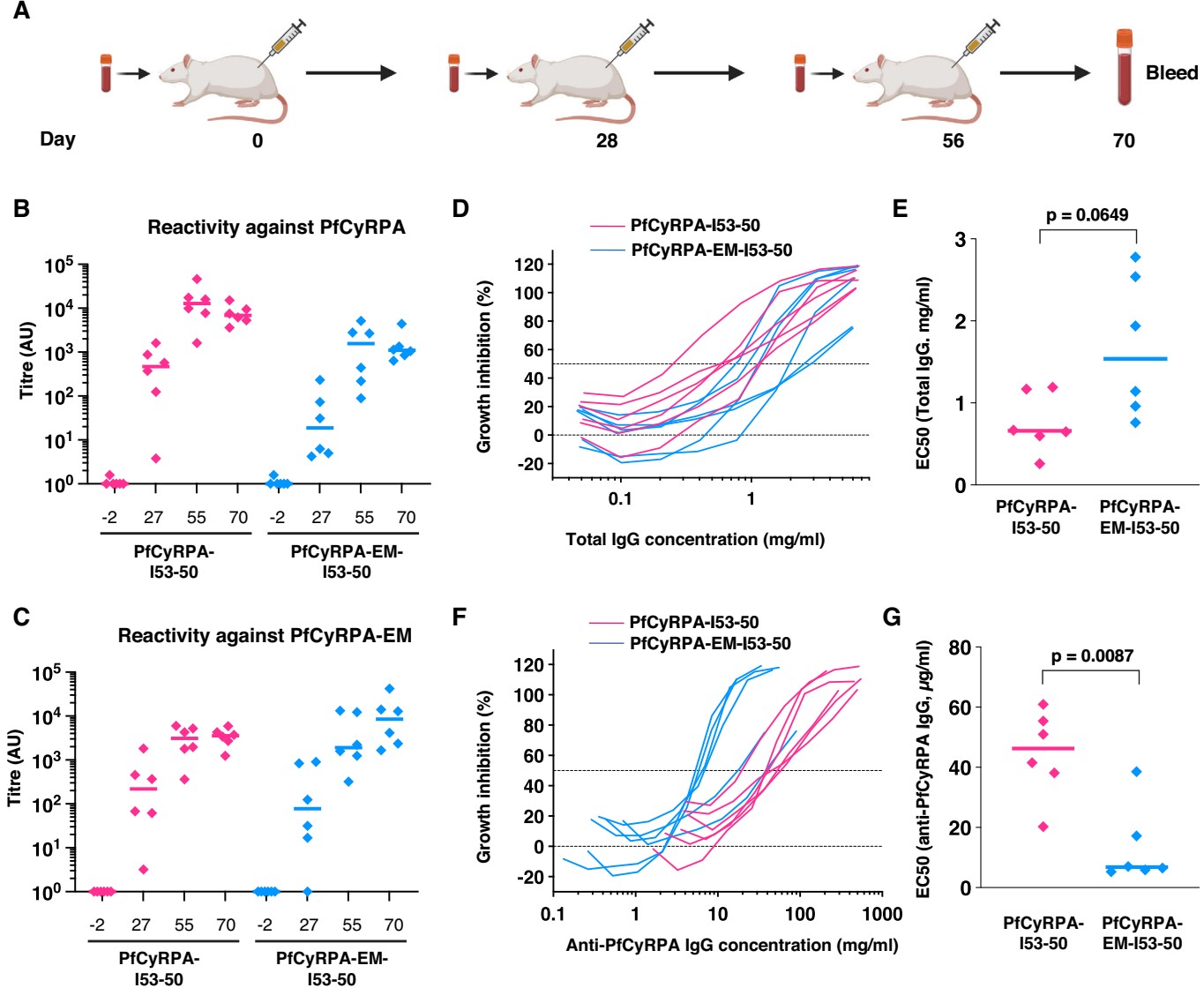

**Figure 3.  Comparison of the immunogenicity of PfCyRPA-I53-50 and PfCyRPA-EM-I53-50.**

(A) Immunisation scheme for rats immunised with PfCyRPA-EM-I53-50 and PfCyRPA-I53-50 formulated with 25 µg of Matrix-M®. Blood samples were taken on days -2, 27, 55, and 70, and immunisations were conducted on days 0, 28, and 56. (B, C) Total IgG purified from rats immunised with PfCyRPA-EM-I53-50 (blue) and PfCyRPA-I53-50 (pink) were assessed for their binding to immobilised (B) PfCyRPA and (C) PfCyRPA-EM by ELISA, with each point representing an individual animal ($n = 6$ per immunogen). (D) Growth-inhibitory activity of different concentrations of total IgG purified from sera raised by immunisation of rats with PfCyRPA-EM-I53-50 (blue) and PfCyRPA-I53-50 (pink), with each line representing an individual animal ($n = 6$ per immunogen). (E) The $EC_{50}$ of IgG from (D). (F) Growth-inhibitory activity of different concentrations of IgG purified from sera raised by immunisation of rats with PfCyRPA-EM-I53-50 (blue) and PfCyRPA-I53-50 (pink), calibrated for the quantity of PfCyRPA-specific IgG, with each point representing an individual animal ($n = 6$ per immunogen). (G) The $EC_{50}$ of IgG from (F). In each group, we immunised six rats and points represent data from individual rats with each line representing an individual animal. Statistical significance was determined using a two-tailed Mann–Whitney test. *P* values are indicated on the plots, and each point is an average of three measurements throughout. Source data are available online for this figure.

1.22% of cells were selected. For rounds 2 through 5, the Cy.004 concentration was lowered to allow a more stringent selection based on binding to Cy.004 and cell surface display. The top 0.91%, 0.96%, 0.3%, and 0.43% of cells, from rounds 2, 3, 4, and 5, respectively, were selected. In round 6, cells were bound to Cy.007 at a concentration of 0.1 µg/ml, and the top 1.02% of cells with the highest cell were selected. In the final round, cells were separately bound to Cy.004 and Cy.007. The top 5.08% (for Cy.004) and top 5.44% (for Cy.007) of cells were enriched based on binding to the respective antibodies and cell surface

display and were pooled for downstream sequencing analysis. Twelve clones showing enhanced binding to both Cy.004 and Cy.007, and high cell surface expression, were selected from the final sorted population and analysed by Sanger sequencing. Three representative variants, BDP-YD1-1, BDP-YD1-2, and BDP-YD1-4 were selected by clustering using CD-HIT (Fu et al, 2012) for expression in Expi293F cells, followed by biophysical assessment and structural elucidation. An additional library without random mutagenesis was generated and analysed similarly.

### Conjugation of PfCyRPA-SpyTag to SpyCatcher-displaying HBsAg nanoparticles

C-terminal SpyTagged PfCyRPA was mixed with SpyCatcher-conjugated HBsAg at 1:1 molar ratio and incubated overnight. The assembled nanoparticles were then purified by size-exclusion chromatography using a Superose 6 Increase 10/300 GL column equilibrated with PBS. The fractions corresponding to the nanoparticle size were pooled and concentrated. The degree of conjugation was assessed using SDS-PAGE. The quality and monodispersity were confirmed DLS.

### In vitro assembly of I53-50-based nanoparticles

Antigen-displaying I53-50 nanoparticles were assembled by mixing trimeric PfCyRPA-I53-50A or PfCyRPA-EM-I53-50A with I53-50B.4PT1 pentamers at an equimolar trimer:pentamer ratio. The mixture was incubated at room temperature for 30 min to allow self-assembly into icosahedral nanoparticles. The assembled nanoparticles were then purified by size-exclusion chromatography using a Superose 6 Increase 10/300 GL column equilibrated with 25 mM Tris pH 7.4, 250 mM NaCl, 5% glycerol for PfCyRPA-I53-50 and 25 mM Tris pH 7.4, 500 mM NaCl, 5% glycerol for PfCyRPA-EM-I53-50. The fractions corresponding to nanoparticles were pooled and concentrated. Assembly quality and monodispersity were confirmed by dynamic light scattering.

### Dynamic light scattering

Dynamic light scattering was used to calculate the hydrodynamic diameter of assembled nanoparticles. Measurements were performed using Viscotek DLS at 20 °C. Nanoparticle samples were diluted to a final concentration of ~0.1 mg/ml and filtered with a 0.1 µm spin filter (Millipore) to remove aggregates. Approximately 20 µl was transferred to a low-volume cuvette for analysis. Each sample was measured 10 times with 10 s per measurement. Increased viscosity due to 5.0% v/v glycerol in the buffer was accounted for during the data analysis.

### Circular dichroism

For circular dichroism, PfCyRPA and PfCyRPA-EM, were buffer-exchanged into 10 mM sodium phosphate pH 7.5, 50 mM NaF using Zeba Spin Desalting Columns (Thermo Fisher Scientific). CD spectra were recorded for PfCyRPA-EM at 10 µM and for PfCyRPA at 4.5 µM, across a wavelength range of 250-190 nm. Measurements were taken every 0.5 nm at 2 °C intervals from 40 °C to 98 °C for PfCyRPA-EM and from 40 °C to 90 °C for PfCyRPA. A baseline measurement for buffer alone at 30 °C was subtracted from all spectra. A Jasco J-815 Spectropolarimeter was used for all measurements. Data analysis was performed using GraphPad Prism v.10.4.1.

### Surface plasmon resonance

Surface plasmon resonance experiments were carried out using a Biacore T200 instrument (Cytiva) with a buffer of 20 mM Tris pH 7.4, 150 mM NaCl, 0.05% Tween-20 and 1 mM CaCl$_2$. Recombinant Protein A/G (Pierce) was coupled to a CM5 chip by amine coupling. Antibody was captured onto the chip through binding to Protein A/G, and the assessed proteins were flowed over at 30 µL/min. The binding of antibodies Cy.003, Cy.004, and Cy.007 to antigen-displaying nanoparticles, PfCyRPA-HBsAg, PfCyRPA-I53-50, and PfCyRPA-EM-I53-50, each at a concentration of 50 nM, was assessed with a contact time of 240 s and a dissociation time of 480 s. Kinetics of binding to antibodies Cy.004 and Cy.007 were measured for PfCyRPA, BDP-YD1-2 and PfCyRPA-EM using twofold serial dilution of analytes, starting at 50 nM for PfCyRPA, with a contact time of 120 s and a dissociation time of 240 s. For the PROSS design, a starting concentration of 250 nM was used. After each cycle, the surface was regenerated using 10 mM using glycine pH 1.75. Data were analysed using the BiaEvaluation software.

### Crystallisation and structure determination

To determine the structure of PfCyRPA-EM bound to Cy.004 Fab, components were combined at a 1:2 molar ratio, incubated for 1 h and complex was purified by size-exclusion chromatography using a Superdex 200 10/200 column (Cytiva) previously equilibrated with 20 mM HEPES pH 7.5, and 150 mM NaCl. Crystals were obtained in sitting drops by vapour diffusion by mixing 100 nl of 8 mg/ml protein with 100 nl well solution (0.1 M Sodium acetate, pH 4.0, 0.2 M Ammonium acetate and 15% w/v PEG 4000). They were cryoprotected by transferring into drops of well solution supplemented with 25% glycerol and cryocooled in liquid nitrogen for data collection. Data were collected at Diamond Light Source at the I04-1 beamline and were indexed using DIALS (v3.0) (Winter et al, 2022) and scaled using AIMLESS (0.7.7) (Evans and Murshudov, 2013), giving a dataset at a resolution of 2.65 Å. The structure was solved using molecular replacement using Phaser MR (1.3) (McCoy et al, 2007), using the fragment of PfCyRPA containing blades 1 and 2, with loops removed and Cy.004 (PDB: 7PHW) (Ragotte et al, 2022) as search models. The model was built and refined using cycles of COOT (v0.8.9.2) (Emsley et al, 2010) and PHENIX (v1.21) (Liebschner et al, 2019).

### Immunisation of rats

Immunisation studies were carried out by Noble Life Sciences (Maryland, USA, which was AALACi accredited and OLAW assured) in full compliance with ethical regulations, using female Wistar IGS rats (n = 6 per group), aged 8–12 weeks and weighing 150–200 g. Animals received intramuscular (IM) injections of vaccine antigen formulated with 25 µg of Matrix-M® adjuvant (Novavax). For the comparison of immunogenicity of PfCyRPA-I53-50 and PfCyRPA-HBsAg, a total 2 µg dose of each was selected. For the comparison of immunogenicity of PfCyRPA-I53-50 and PfCyRPA-EM-I53-50, a 2 µg dose of PfCyRPA-I53-50 and an equimolar amount of 1.4 µg PfCyRA-EM-I53-50 was selected. Group sizes were consistent with prior studies using similar immunisation protocols. Blood was collected by tail vein or submandibular bleed on days −2, 27, and 55 to obtain serum. Final blood samples were collected by terminal cardiac bleed on day 70. All sera were frozen and transported to the University of Oxford, UK, for further analysis. Each rat was evaluated individually using ELISA and a growth inhibition assay. No blinding was applied during the study, as assays resulted in quantitative data from bulk samples.

### ELISA measurements

Nunc Maxisorp plates (Thermo Fisher Scientific) were coated overnight at 4 °C ( > 16 h) with 50 µl per well of PfCyRPA or PfCyRPA-EM at a concentration of 2 µg/ml, diluted in Dulbecco's PBS (DPBS). Following coating, plates were washed six times with PBS containing 0.05% Tween-20 (PBST) and then blocked with

200 µl per well of Starting Block T20 (Thermo Fisher Scientific) for 1 h at room temperature. After blocking, the plates were again washed six times with PBST. For anti-PfCyRPA ELISAs the reference serum had previously been raised in rats immunised with three doses of 20 µg PfCyRPA formulated with 25 µg of Matrix-M®. For anti-PfCyRPA-EM ELISA, a pooled serum from PfCyRA-EM-I53-50-immunised rats from the current study was used as the reference serum. Test serum samples and reference serum were diluted in Starting Block T20 and added to the plates at 50 µl per well. Plates were incubated at room temperature for 2 h. Following serum incubation, plates were washed six more times with PBST, and 50 µl per well of goat anti-rat IgG secondary antibody (Sigma, A8438; 1:1000 dilution in Starting Block T20) was added. Plates were incubated for 1 h at room temperature and washed six times with PBST. Colorimetric detection was performed by adding 100 µl per well of p-nitrophenyl phosphate substrate in diethanolamine buffer. The reaction was developed until internal control wells reached an OD405 of ~1.0 (typically within the range of 0.8–1.2). Each serum sample was tested in triplicate against each antigen coating. Sample dilutions were selected to ensure OD readings fell within the linear range of the standard curve. To convert OD values into antigen-specific IgG concentrations, a conversion factor was applied based on concentration-free calibration analysis, as previously described (Williams et al, 2024).

### IgG purification from rat serum

Serum samples (3–4 ml) were diluted with IgG binding buffer (Thermo Scientific) to a total volume of 8–10 ml and applied to a 2 ml protein G column (Immunopure Plus, Pierce) pre-equilibrated with 10 ml of binding buffer. The initial flow-through was reapplied to the column three additional times to maximise IgG capture. The column was washed twice with 10 ml binding buffer, and bound IgG was eluted using 10 ml of 0.1 M glycine (pH 2.7), collected directly into 350 µl of 1 M Tris buffer (pH 9.0) to neutralise the eluate. Eluted IgG was buffer-exchanged first into RPMI 1640 (Sigma) using a 30 kDa molecular weight cut-off Amicon Ultra centrifugal filter (15 ml, Fisher Scientific), followed by exchange into incomplete medium, ICM (RPMI supplemented with 1% L-glutamine, 0.005% hypoxanthine, and 25 mM HEPES, and 0.2% D-glucose), with a final volume of approximately 250 µl. To prepare IgG samples for GIA, they were clarified by adding 50 µl of 100% haematocrit human O+ red blood cells (RBCs) for each 1 ml of the original serum sample that was used to prepare the IgG. Samples were gently inverted for 1 h at room temperature to adsorb any anti-human RBC IgG. Following incubation, RBCs were pelleted by centrifugation at 5000×g for 2 min. The clarified IgG-containing supernatants were filtered through a 0.22-µm Spin-X filter (Costar, Scientific Laboratory Supplies), and the total IgG concentration was measured by absorbance at 280 nm. To determine the concentration of PfCyRPA-specific IgG, ELISA was performed using PfCyRPA-coated plates. ELISA absorbance values were converted to concentration using a calibration factor derived from a previously established concentration-free standard curve (Williams et al, 2024).

### Growth-inhibitory activity measurement

Red blood cells used for parasite culture and GIA were obtained from the NHS Blood and Transplant service, with ethical approval from the University of Oxford Medical Sciences Inter-Divisional research ethics committee. No personal information was collected. Parasite cultures were grown in media supplemented with human serum from O+

donors for the CyRPA-HBsAg vs CyRPA-I53-50 comparison study in Fig. 1, whereas AlbuMAX™ (Thermo Fisher Scientific) was used for the CyRPA-I53-50 vs PfCyRPA-EM-I53-50 comparison study in Fig. 3. For the human serum supplemented media, complete medium was composed of RPMI containing 1% L-glutamine, 0.005% hypoxanthine, 25 mM HEPES, 10 µg/ml gentamycin (Sigma) and 10% heat-inactivated pooled human serum from O+ donors, whereas 0.5% AlbuMAX™ with the addition of 0.2% D-glucose was added in the AlbuMAX™ supplemented medium. The respective 2× complete medium contained 20% heat-inactivated pooled human serum from O+ donors or 1.0% AlbuMAX™ with the addition of 0.2% D-glucose. On the first day of the assay, P. falciparum 3D7 culture at 2% haematocrit in complete medium was synchronised at 37 °C using 5% sorbitol. On day 2, synchronised parasites were diluted to 0.4% parasitaemia in 2× complete medium at 2% haematocrit. A volume of 20 µl of this parasite suspension was added to each well of a 96-well plate, already containing 20 µl of 10 mM EDTA, incomplete medium, purified rat IgG twofold serial dilutions, positive control antibodies (40 µg/ml 2AC7, 30 µg/ml R5.016, or 5 µg/ml R5.034, each of which binds PfRH5), or a negative control monoclonal antibody (1000 µg/ml DB8, which binds PvDBP). A parallel tracker culture with 0.4% final parasitaemia was also prepared. Assay plates and tracker cultures were incubated at 37 °C in a modular incubator for 44 h. On day 4, plates were washed twice with 100 µl cold PBS per well, and erythrocytes were resuspended by shaking at 1300 rpm for one minute (Titramax 100; Heidolph). Parasite growth was quantified using a lactate dehydrogenase (LDH) assay. A total of 120 µl of LDH substrate was added to each well. The substrate contained 3-acetylpyridine adenine dinucleotide (50 µg/ml, Sigma), diaphorase (1 U/ml, Sigma), and nitro

---

**The paper explained**

**Problem**

Malaria causes over 600 thousand deaths and hundreds of millions of cases each year, and we urgently require new vaccines. The parasites that cause malaria replicate within human blood cells. If we can stop red blood cell invasion, we can prevent the symptoms and transmission of the disease. The PfPCRCR complex, consisting of five protein components, is required for invasion, and its PfRH5 component is a leading blood-stage malaria vaccine, currently in clinical trials. However, while PfRH5 induces some protection in humans, this is insufficient and incomplete. Can we design vaccine immunogens based on other components of PfPCRCR that induce growth-inhibitory responses, and which we can include in future blood-stage malaria vaccines?

**Results**

Here, we studied the PfCyRPA component of PfPCRCR. Previous studies of PfCyRPA revealed it to form a structure known as a β-propeller, which has six 'blades' arranged around a circle. Studies of PfCyRPA bound to monoclonal antibodies showed that all known growth-inhibitory antibodies bind to two of these blades. We therefore used rational structure-guided methods to design a novel protein that consists of only these two blades of PfCyRPA, which we call PfCyRPA-EM. We found that PfCyRPA-EM is more stable and easier to produce than PfCyRPA and, in a pre-clinical model, it induces a high-quality immune response that prevents parasite growth.

**Impact**

We have designed a novel synthetic malaria vaccine immunogen, based on PfCyRPA, which induces a high-quality parasite growth-inhibitory response. This is now available for clinical testing, either alone or in combination with vaccine components.

blue tetrazolium (0.2 mg/ml, Sigma). Plates were read at 650 nm, targeting an $OD_{650}$ of 0.4–0.6 in the no-inhibition ICM control wells. Percent growth inhibition was calculated by defining the EDTA control as 100% inhibition and the ICM control as 0% inhibition:

$$\% \text{ inhibition} = 100 - \left(\frac{(\text{OD650 test sample} - \text{OD650 EDTA control})}{(\text{OD650 ICM control} - \text{OD650 EDTA control})}\right) \times 100\%$$

## Quantification and statistical analysis

All data were analysed using GraphPad Prism (v10.4.1) (GraphPad Software Inc., CA, USA). Tests and statistics are described in the figure legends. To determine $EC_{50}$ values, mAb dilution curves were fitted to a curve by four-parameter non-linear regression. GIA values were interpolated from the resultant curve with upper and lower 95% confidence intervals. A $P$ value of less than 0.05 was considered statistically significant. Bio-Rad S3e™ ProSort™ and FlowJo v10.10 were employed for cell sorting data visualisation and analysis. PyMOL was used for molecular visualisation and image generation. No blinding was conducted as outputs were quantitative. No samples were excluded from analysis.

## Data availability

The crystal structure is available at the Protein Data Bank with accession code 9OSX.

The source data of this paper are collected in the following database record: biostudies:S-SCDT-10_1038-S44321-026-00376-x.

## Peer review information

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

## Acknowledgements

This work was funded by a Wellcome Investigator Award (220797/Z/20/Z) to MKH and the Infectious Disease Division, Bureau for Global Health, United States Agency for International Development (USAID), under the terms of contract 7200AA20C00017 with PATH. Simon Draper and colleagues at PATH (Ashley Birkett, Randall MacGill, and Allison Clifford) and USAID (Lorraine Soisson and Robin Miller) contributed to discussions about this project and to project management activities. The opinions expressed herein are those of the authors and do not necessarily reflect the views of USAID. The authors thank Sumi Biswas for the provision of the HBsAg virus-like particle, Ed Lowe and the beamline scientists at Diamond Light Source for support with crystallographic data collection and David Staunton for help with biophysical analysis. NHS Blood & Transplant (NHSBT) have provided material, but the views expressed are not necessarily those of NHSBT.

## Author contributions

**Nawsad Alam**: Conceptualisation; Formal analysis; Validation; Investigation; Visualisation; Methodology; Writing—original draft; Writing—review and editing. **Clare Wolfle**: Conceptualisation; Formal analysis; Investigation; Visualisation. **Egle Butkeviciute**: Formal analysis; Investigation. **Doris Quinkert**: Investigation. **Lloyd D W King**: Investigation. **Matthew K Higgins**: Conceptualisation; Formal analysis; Supervision; Funding acquisition; Validation; Investigation; Visualisation; Methodology; Writing—original draft; Writing—review and editing.

Source data underlying figure panels in this paper may have individual authorship assigned. Where available, figure panel/source data authorship is listed in the following database record: biostudies:S-SCDT-10_1038-S44321-026-00376-x.

## Disclosure and competing interests statement

NA and MKH are named inventors on patent applications relating to PfCyRPA-EM and/or other malaria vaccines, and immunisation regimes. NA, MKH, LDWK, and DQ are inventors on patent applications relating to RH5 malaria vaccines and/or antibodies.

# Expanded View Figures

**Figure EV1.** **Assembly and biophysical characterisation of PfCyRPA-I53-50 and PfCyRPA-HBsAg nanoparticles.**

(**A**) Size-exclusion chromatography traces and SDS-PAGE for the individual components required for nanoparticle assembly (representative from $n = 2$). (**B**) The top panel illustrates assembly of PfCyRPA-HBsAg nanoparticles from PfCyRPA with a c-terminal Spy tag (PfCyRPA-cSpyTag) and HBsAg conjugated to SpyCatcher (SpyCatcher-HBsAg). The central panel illustrates assembly of PfCyRPA-I53-50 from I53-50B.4PT1 and PfCyRPA-I53-50A. The lower panel shows size-exclusion chromatography traces and SDS-PAGE for the assembled PfCyRPA-I53-50 and PfCyRPA-HBsAg nanoparticles. For each trace, the fractions selected for biophysical characterisation and immunisation are shaded (representative from $n = 2$). (**C**) Dynamic light scattering traces for PfCyRPA-I53-50 (top) and PfCyRPA-HBsAg (bottom) (representative from $n = 2$). (**D**) Surface plasmon resonance traces for PfCyRPA-I53-50 (top) and PfCyRPA-HBsAg (bottom) nanoparticle binding to Cy.003 (orange), Cy.004 (purple), and Cy.007 (green). The analyte binding level for each run was normalised relative to the antibody capture for that run; representative from $n = 2$.

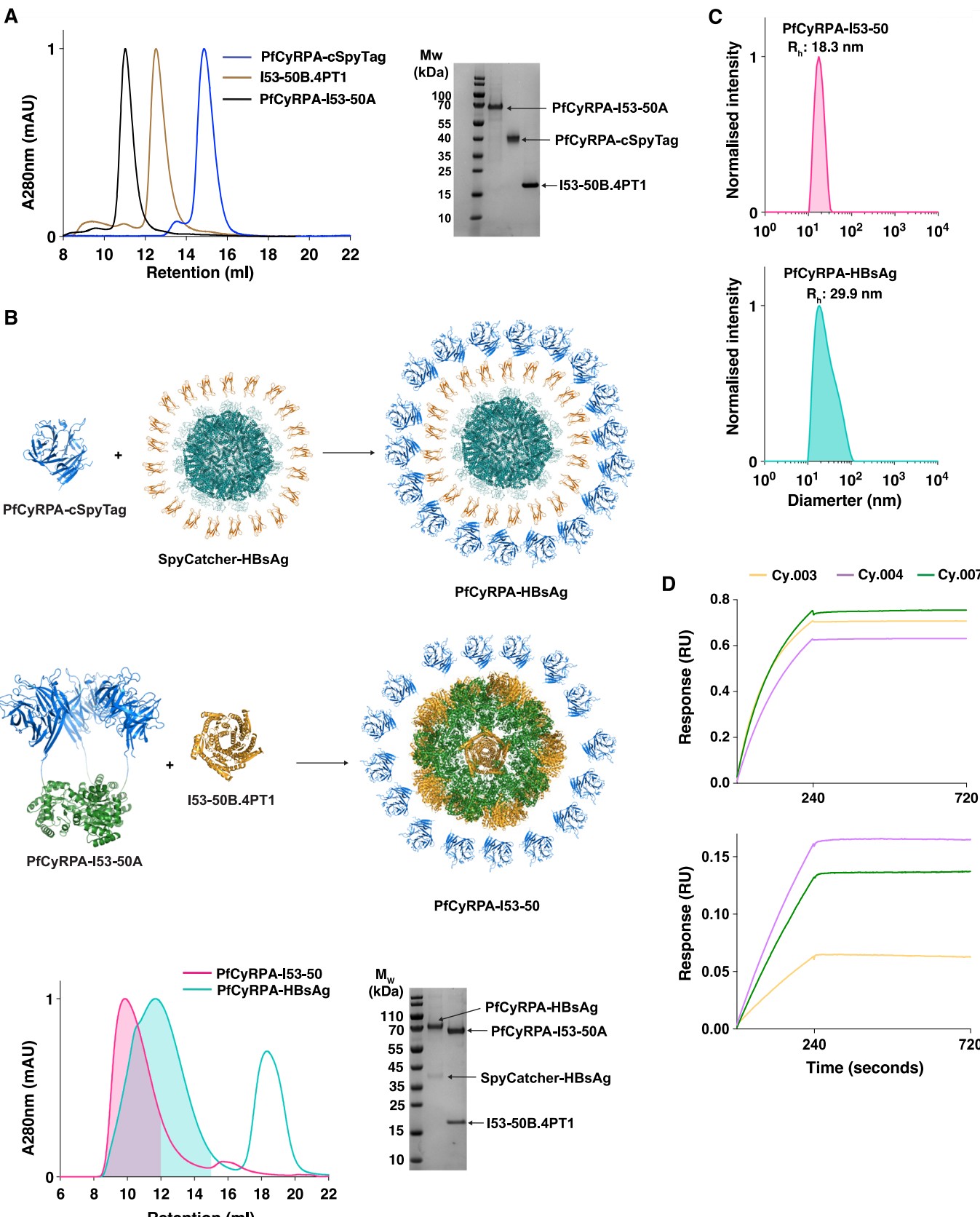

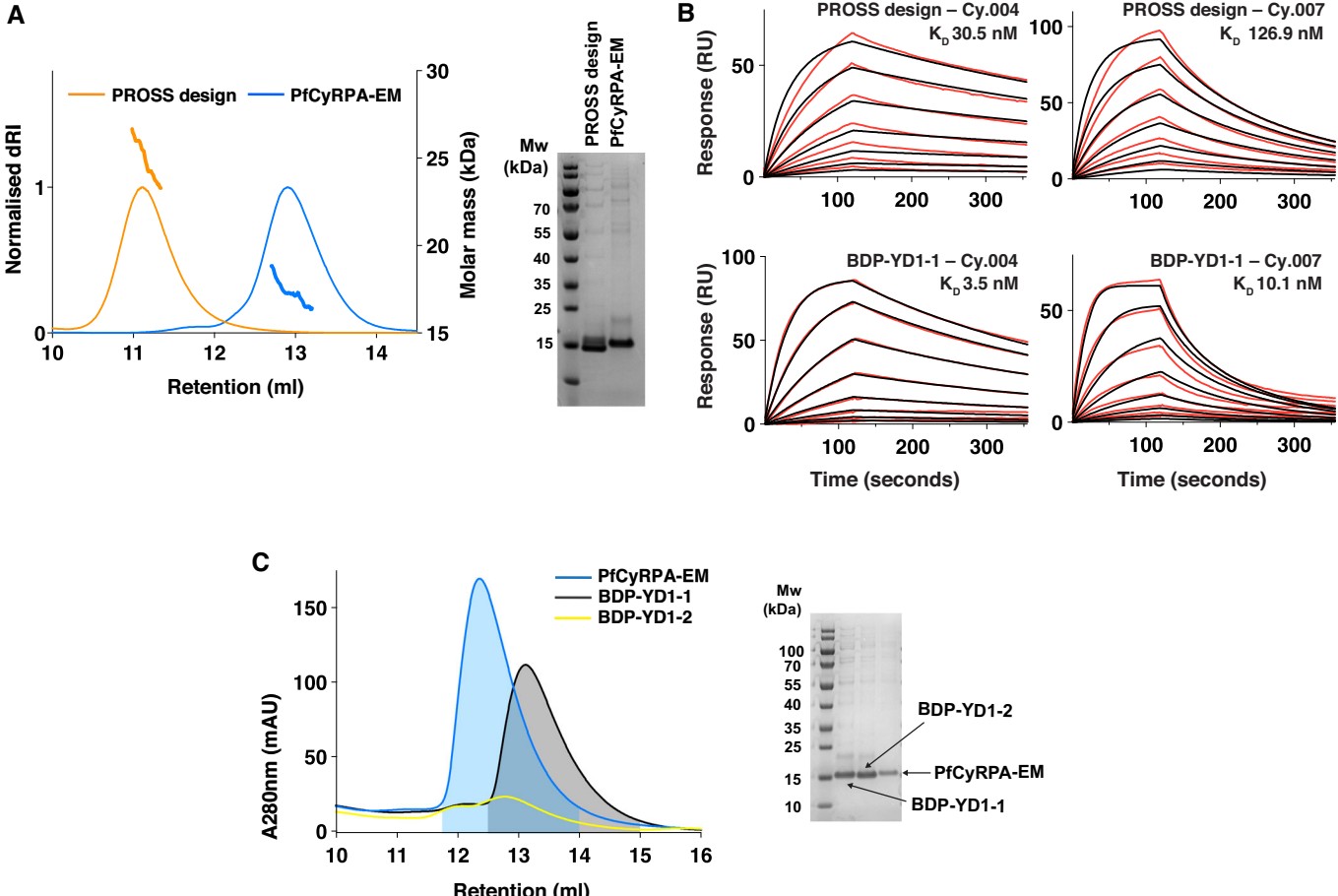

**Figure EV2.  Purification and biophysical characterisation of the PROSS design and yeast display derived designs.**

(**A**) SEC-MALLS traces and SDS-PAGE gel for the PROSS design and PfCyRPA-EM. (**B**) Surface plasmon resonance traces for PROSS design (top) and BDP-YD1-1 (bottom) binding to Cy.004 (left) and Cy.007 (right), with the data shown as black lines. The data was fitted to a 1-to-1 binding model for BDP-YD-1 and a bivalent analyte model for PROSS design, with fits shown as red lines (representative from $n = 2$). (**C**) Size-exclusion chromatography traces and SDS-PAGE for the designs selected after yeast display (representative from $n = 2$).

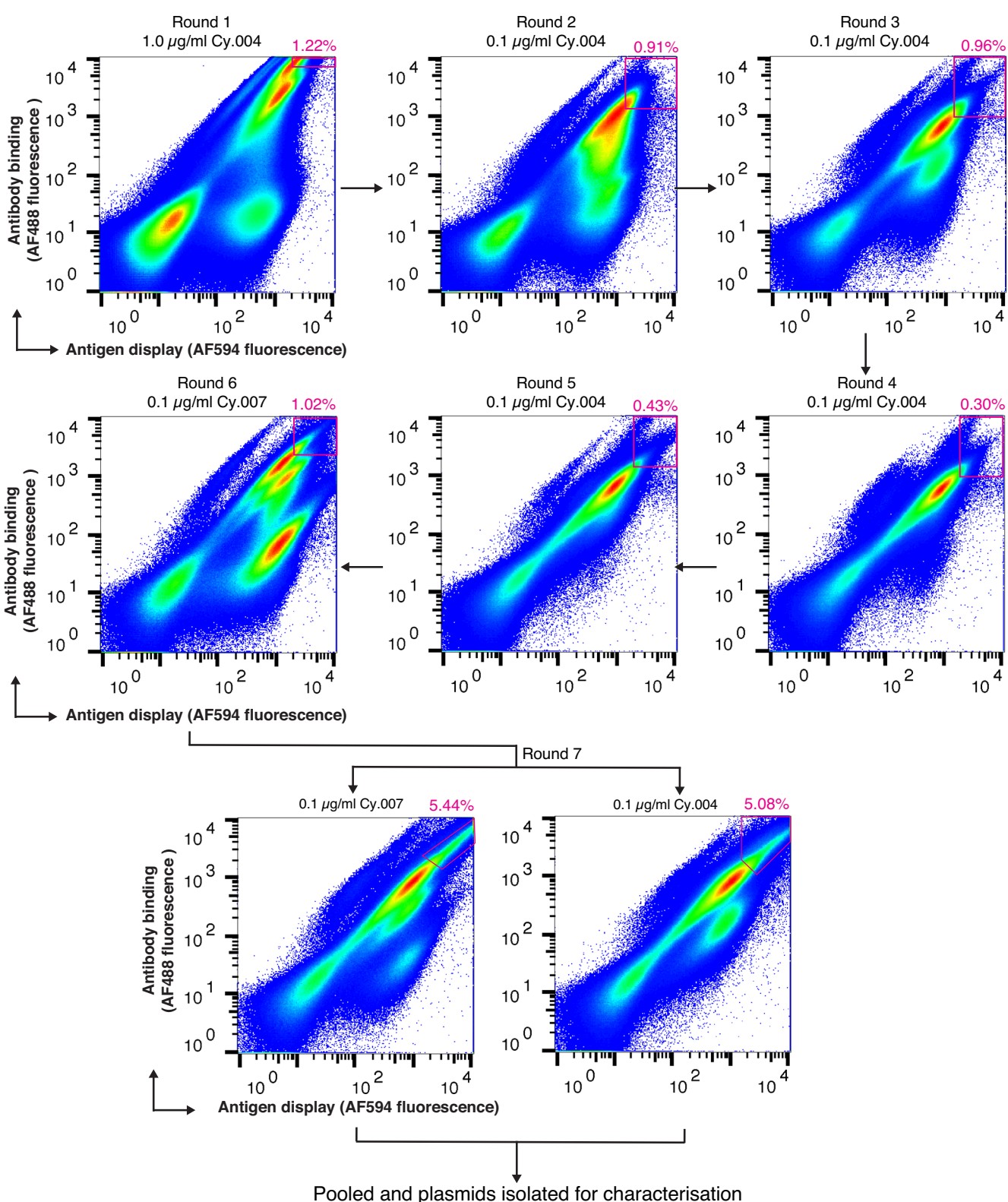

**Figure EV3. Yeast display library sorting by flow cytometry.**

Yeast cells were double labelled to sort based on both expression (X-axis, AlexaFluor594 fluorescence) and antigen binding (Y-axis, AlexaFluor488 fluorescence). For each sorting round, the name and concentration of the antigen binding antibody used is indicated on the top of each panel. The cells within the gates, delineated with magenta lines and with percentages shown, were selected for the next round.

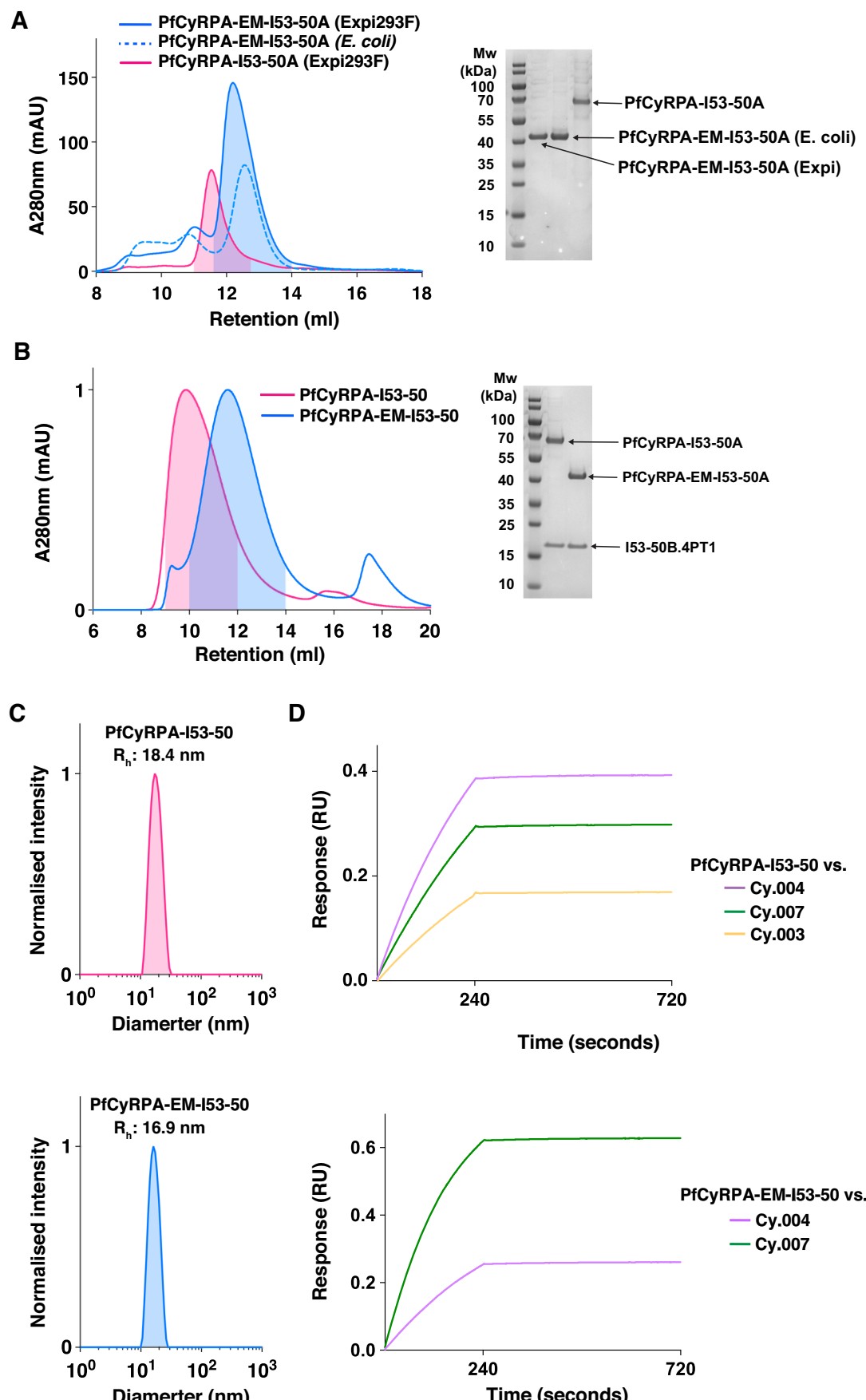

**Figure EV4. Assembly and biophysical characterisation of PfCyRPA-EM-I53-50 and PfCyRPA-I53-50.**

(A) Purification of components of the I53-50 fused nanoparticles, shown on SEC traces (left) and an SDS-PAGE gel (right). PfCyRPA-I53-50A and PfCyRPA-EM-I53-50A were purified from Expi293F cells and PfCyRPA-EM-I53-50A was purified from *E. coli* (representative from $n = 2$). (B) Size-exclusion chromatography traces obtained for assembled PfCyRPA-I53-50 and PfCyRPA-EM-I53-50 particles (representative from $n = 2$). The fraction selected for biophysical characterisation and immunisation is shaded. (C) Dynamic light scattering traces for PfCyRPA-I53-50 (top) and PfCyRPA-EM-I53-50 (bottom). (D) Surface plasmon resonance traces for PfCyRPA-EM-I53-50 binding (top) to Cy.003 (orange), Cy.004 (pink), and Cy.007 (green) and for PfCyRPA-EM-I53-50 binding to Cy.004 (pink) and Cy.007 (green) (bottom). The analyte binding level for each run was normalised relative to the antibody capture for that run (representative from $n = 2$).

