## [Peer Review File · EMBO Molecular Medicine]

Structure-guided design of a PfCyRPA-based vaccine against blood-stage malaria

Nawsad Alam, Clare Wolfle, Egle Butkeviciute, Doris Quinkert, Lloyd King, and Matthew Higgins

Corresponding author: Matthew Higgins (matthew.higgins@bioch.ox.ac.uk)

Review Timeline:

Submission Date:	21st Jun 25
Editorial Decision:	28th Jul 25
Revision Received:	27th Nov 25
Editorial Decision:	17th Dec 25
Revision Received:	9th Jan 26
Accepted:	14th Jan 26

Editor: Lise Roth

Transaction Report:

28th Jul 2025

Dear Dr. Higgins,

Thank you for submitting your manuscript to EMBO Molecular Medicine and please accept my apologies for the delay in getting back to you as we were waiting for a referee report. Unfortunately, referee #1 has not yet got back to us, but given that both referees #2 and #3 make similar recommendations, we prefer to make a decision now to avoid further delay in the process. Should referee #1 provide a report, we will send it to you on the understanding that we will not ask you to carry out any extensive experiments beyond those requested in the attached reports from referees #2 and #3. As you will see from the reports below, the referees recognize the interest of the study and generally support publication of your work, subject to appropriate revisions.

Addressing the reviewers' concerns in full will be necessary for further considering the manuscript in our journal, and acceptance of the manuscript will entail a second round of review. EMBO Molecular Medicine encourages a single round of revision only and therefore, acceptance or rejection of the manuscript will depend on the completeness of your responses included in the next, final version of the manuscript. For this reason, and to save you frustration at the end, I would strongly discourage you from returning an incomplete revision.

We are expecting your revised manuscript within three to four months, if you anticipate any delay, please contact us.

We require:

4) A .docx formatted letter INCLUDING the reviewers' reports and your detailed point-by-point responses to their comments. As part of the EMBO Press transparent editorial process, the point-by-point response is part of the Review Process File (RPF), which will be published alongside your paper.

5) A complete author checklist, which you can download from our author guidelines (<https://www.embopress.org/page/journal/17574684/authorguide#submissionofrevisions>). Please insert information in the checklist that is also reflected in the manuscript. The completed author checklist will also be part of the RPF.

6) All Materials and Methods need to be described in the main text using our 'Structured Methods' format. According to this format, the Methods section includes a Reagents and Tools Table (listing key reagents, experimental models, software and relevant equipment and including their sources and relevant identifiers) followed by a Methods and Protocols section describing the methods, ideally using a step-by-step protocol format. The aim is to facilitate adoption of the methodologies across labs. Please download and fill our Reagents and Tools Table template (.docx), which you can find in our author guidelines:

<https://www.embopress.org/doi/10.15252/msb.20178071>

7) Please note that all corresponding authors are required to supply an ORCID ID for their name upon submission of a revised manuscript.

8) It is mandatory to include a 'Data Availability' section after the Materials and Methods. Before submitting your revision, primary datasets produced in this study need to be deposited in an appropriate public database, and the accession numbers and

database listed under 'Data Availability'. Please remember to provide a reviewer password if the datasets are not yet public (see <https://www.embopress.org/page/journal/17574684/authorguide#dataavailability>).

9) For data quantification: please specify the name of the statistical test used to generate error bars and P values, the number (n) of independent experiments (specify technical or biological replicates) underlying each data point and the test used to calculate p-values in each figure legend. The figure legends should contain a basic description of n, P and the test applied. Graphs must include a description of the bars and the error bars (s.d., s.e.m.). Please provide exact p values.

10) Our journal encourages inclusion of *data citations in the reference list* to directly cite datasets that were re-used and obtained from public databases. Data citations in the article text are distinct from normal bibliographical citations and should directly link to the database records from which the data can be accessed. In the main text, data citations are formatted as follows: "Data ref: Smith et al, 2001" or "Data ref: NCBI Sequence Read Archive PRJNA342805, 2017". In the Reference list, data citations must be labeled with "[DATASET]". A data reference must provide the database name, accession number/identifiers and a resolvable link to the landing page from which the data can be accessed at the end of the reference. Further instructions are available at .

11) We replaced Supplementary Information with Expanded View (EV) Figures and Tables that are collapsible/expandable online. EV Figures should be cited as 'Figure EV1, Figure EV2' etc... in the text and their respective legends should be included in the main text after the legends of regular figures.

12) The paper explained: EMBO Molecular Medicine articles are accompanied by a summary of the articles to emphasize the major findings in the paper and their medical implications for the non-specialist reader. Please provide a draft summary of your article highlighting

13) Author contributions: CRediT has replaced the traditional author contributions section because it offers a systematic machine readable author contributions format that allows for more effective research assessment. Please remove the Authors Contributions from the manuscript and use the free text boxes beneath each contributing author's name in our system to add specific details on the author's contribution. More information is available in our guide to authors.

Please also suggest a visual abstract to illustrate your article as a PNG file 550 px wide x 300-600 px high. A cropped portion of this image will serve as thumbnail for the table of content on our webpage.

16) As part of the EMBO Publications transparent editorial process initiative (see our Editorial at <http://embomolmed.embopress.org/content/2/9/329>), EMBO Molecular Medicine will publish online a Review Process File (RPF) to accompany accepted manuscripts.

In the event of acceptance, this file will be published in conjunction with your paper and will include the anonymous referee reports, your point-by-point response and all pertinent correspondence relating to the manuscript. Let us know whether you

agree with the publication of the RPF and as here, if you want to remove or not any figures from it prior to publication. Please note that the Authors checklist will be published at the end of the RPF.

I look forward to receiving your revised manuscript.

Yours sincerely,

Lise Roth

***** Reviewer's comments *****

Referee #2 (Comments on Novelty/Model System for Author):

This manuscript reports on the rational design and optimization of a blood-stage malaria vaccine targeting the *Plasmodium falciparum* cysteine-rich protective antigen (PfCyRPA), a key component of the PfPCRCR complex. The PCRCR complex is required for red blood cell invasion and includes Rh5 protein that is a target for a current blood stage vaccine Rh5. To enhance immunogenicity, authors compared two nanoparticle display systems for presenting PfCyRPA: a post-production conjugation approach and a direct fusion strategy using the I53-50 nanoparticle scaffold. The direct fusion system proved superior, yielding higher titers of growth-inhibitory antibodies, improved manufacturability, and greater consistency. For improved antibody quality, authors applied structure-guided immunogen design. They engineered a synthetic variant, PfCyRPA-EM, which focuses the immune response on the most protective epitopes (blades 1 and 2 of the β -propeller domain). By leveraging computational protein design (PROSS) and yeast display selection for growth-inhibitory antibody binding, they obtained a correctly folded, highly stable immunogen. PfCyRPA-EM demonstrated a 14^{°C} increase in thermal stability, six-fold higher expression in mammalian cells, and compatibility with bacterial expression as an I53-50 fusion. In a preclinical rat model, PfCyRPA-EM-I53-50 demonstrated improved quality as it was induced more efficacious parasite growth-inhibitory antibodies than the original PfCyRPA-I53-50. The enhanced stability and manufacturability of PfCyRPA-EM further highlighted its suitability for vaccine development and distribution. Given the partial and non-durable protection of current leading candidates (e.g., RH5.1), the new PfCyRPA-EM immunogen represents a significant advance. It is now positioned for combination studies with other PfPCRCR components, with the aim of achieving more comprehensive and durable protection against malaria.

Key Innovations:

- Nanoparticle display optimization: Direct fusion to I53-50 for superior immunogenicity and manufacturability.
- Structure-based immunogen design: Focused epitope presentation for higher quality antibody responses.
- Enhanced manufacturability: Improved stability and expression in both mammalian and bacterial systems.
- Translational potential: PfCyRPA-EM is a strong candidate for next-generation malaria vaccines, particularly in multicomponent formulations.

This report exemplifies how advanced protein engineering and immunogen design can address both immunological and practical challenges in vaccine development.

Referee #2 (Remarks for Author):

The manuscript is an excellent example of how advanced protein engineering and immunogen design can address both immunological and practical challenges in vaccine development. It is of high technical quality and employs cutting-edge

approaches in protein design. My suggestions are minor and intended to further improve clarity and accessibility for nonspecialist readers.

1. Introduction: The authors provide a compelling rationale against using full-length proteins for immunogen design, noting that such proteins often contain epitopes that elicit non-functional antibodies, thereby diminishing the quality of the immune response. They also reference studies showing that the most inhibitory antibodies against PfCyRPA are directed against blades 1 and 2 of the β -propeller domain. However, it is surprising that the initial Results section presents experiments using full-length PfCyRPA. I recommend revising the Introduction to include additional background on the nanoparticle platforms used, specifically describing the properties of HBsAg and I53-50. The differences between these two particles are not immediately apparent to nonspecialists, making it difficult to appreciate the methodological rationale. For example, how many immunogens are displayed per particle? Should this difference influence the choice of formulation dose? Additionally, the rationale for removing potential PfCyRPA glycosylation sites is unclear and should be clarified.

2. Discussion: This section largely reiterates the results rather than interpreting them. It would benefit from a more in-depth discussion of the key differences between the two nanoparticle platforms tested. For instance, is the observed inferiority of HBsAg attributable to conjugation efficiency or other intrinsic properties of the protein? What types of nanoparticles are currently available for vaccine development? Finally, it remains unclear how the efficacy of PfCyRPA-EM-I53-50 compares to the most potent version of Rh5. Is this new particle a significant advancement towards a highly effective blood-stage malaria vaccine, or is it better considered as a complementary component to Rh5?

Referee #3 (Comments on Novelty/Model System for Author):

Experimental reproducibility of some of the work (ELISA's, Growth assays) is a bit unclear. Since the conclusions are based on these assay readouts for one vaccine platform over another, it would be better to have stronger comparative data. The improved vaccine platform performance is a real positive. It would be good to explore whether two immunisations would be enough to generate a promising growth inhibitory immune response with the optimised vaccine. Medical impact will strengthen when used in combination with Rh5 antibodies, which are not done here.

Referee #3 (Remarks for Author):

In this paper Alam et al. develop an immunogen against blood stage *Plasmodium falciparum* malaria based on PfCyRPA, part of a complex essential for erythrocyte invasion. The authors focus on blades 1 and 2 of PfCyRPA that previous studies have shown to be most immunogenic. Firstly, the paper compares two platforms for antigen presentation and show that direct fusion to I53-50 nanoparticles is simpler to produce and most immunogenic in an in-vitro parasite growth inhibition assay. The study proceeds to a structure guided design of blades 1 and 2 of PfCyRPA in silico to improve epitope folding and expression, however the resulting design forms a dimer and has a lower affinity to known PfCyRPA binding antibodies Cy.004 and Cy007. To improve this design, the authors then generate a yeast surface display library and select for binders of both Cy.004 and Cy.007 using MACS and several rounds of FACS. The resulting PfCyRPA-EM closely resembles the structure of PfCyRPA blades 1 and 2, but has a higher thermal stability than PfCyRPA. The optimised PfCyRPA-EM antigen is finally fused to I53-50 nanoparticles and tested in rats. Compared to PfCyRPA, PfCyRPA-EM induced a lower but more specific anti-PfCyRPA response, resulting in similar parasite growth inhibition.

Overall, this paper presents an improved design and more cost-effective production method of an antigen that structurally mimics blades 1 and 2 of PfCyRPA. The immunogenicity in rats is lower but more specific compared to PfCyRPA. The optimised antigen has the potential to have an additive effect in combination with other antigens to produce an invasion-blocking vaccine against *P. falciparum* malaria.

Specific comments regarding yeast surface display:

- Figure 1 is titled 'Assembly, biophysical characterisation, and immunogenicity.....'. However, there really is not indication of the Assembly process, although it would be quite useful to run the reader through this. I would either change the title or, better yet, add in a diagram of the Assembly.
- In Figures 1, 3, 4, there is no indication of the number of repeat experiments (biological replicates) for assays such as growth assays and ELISAs. Although multiple samples are tested, it is not clear whether the data is from a single experiment or mean/median of multiple experiments. These are not difficult experiments and it is always best to do biological replicates in case the assays are impacted by problems in parasite growth, setup, readout, etc. Biological replicates strengthen the data and interpretability. Since there are no error bars, it seems likely that these experiments were done only once. Since there are outliers, it is not clear whether this represents less efficient antibody production after immunisation with a specific rat or some sort of problem with the assay for specific samples. It is reasonable for replicates to be done for these experiments if they have not been and the combined data shown. Failing that, the number of repeats should be clearly stated so the reader can decide on the strength of the data.
- Please show the gating strategy in Supplementary Figure 3. As presented it appears that the total population from each round is fed into the next round, which contrasts with the figure legend that suggests less than 1% of cells were selected to the next

round.

- In the methods it is unclear what happens in between each sorting round. Please add detail.
- Was there a rationale for starting the selection with Cy.004 rather than Cy.007?
- Figure 2C does not show convincing selection of higher affinity binding. In fact, there is a population with high expression/low binding that appears to be enriched. As shown in Supplementary Figure 3 this non-binding population is lost during Cy.004 selecting rounds but re-appears in round 6 after switching to selection with Cy.007. Perhaps you could graph the percent of cells in the high expression/high binding gate in the non-mutagenized library, the mutagenized library pre-selection and the mutagenized library post selection? A more convincing illustration of the enrichment would be a contrast between the mutagenized library pre and post selection.
- Can the authors discuss the implications for a larger fraction of growth inhibitory antibodies targeting CyRPA for the PfCyRPA-EM-I53-50 immunised samples, even as higher non-inhibitory antibody titres overall mean that the growth inhibitory activity is lower when compared as a proportion of total IgG? Is there a way that these could be selected for to improve CyRPA inhibitory antibody concentration through immunisation? What happens if a bleed post second immunisation is tested, do we see equivalent or proportionally improved CyRPA growth inhibitory antibody levels relative to total IgG or CyRPA antibody activity, even though it does not reach the peak seen after the 3rd immunisation? It would be good to do this. If it was the case that the 2nd immunisation provides the best CyRPA protective response, could immunisation be optimised around 2 vaccinations in preference to 3 where more non-specific/inhibitory IgG seems to increase, but not CyRPA growth inhibitory antibodies.
- The immunogenicity of PfCyRPA-EM is qualitatively similar to that reported by Bjornsson et al. (2024) i.e. immunofocussing of neutralizing antibodies at the expense of overall anti-PfCyRPA antibody titres. The authors mention the limitations of the design of the Bjornsson et al. (2024) study in the introduction, however a comparison of the results between these two studies in the discussion would clarify the impact of the findings of this paper.

Minor specific comments:

L169 - This paragraph has very frequent references to supplementary figures and compares data between the main Figure 2 and supplementary figures. Suggest incorporating some of the supplementary data in Figure 2.

L172 and L173 - Supplementary Figure 2C and Figure 2C labels refer to PfCyRPA-EM, not BDP-YD1-4, even though new nomenclature hasn't been introduced yet. Supplementary Table 2 refers to Yeast display design 3 / BDP-YD1-4 / PfCyRPA-EM. It would be clearer to give the antigen one name throughout.

L190: Missing word I53-50A.1NT1 was expressed

L191: Is the cost effectiveness due to higher expression in Expi293F cells or due to the ability to express in E. coli? As phrased, this is unclear.

L175 : Although much lower than for the PROSS design, the KD of PfCyRPA-EM vs Cy.007 is still approx. 3x higher than the KD for PfCyRPA, meaning there is lower affinity. Do you think this suggests some misfolding?

L192: Missing word: I53-50A.1NT1 was assembled

Figure 4D: Cy.003 binding is missing in bottom panel.

L213: Suggest rephrase to 'there was no statistical difference between the mean EC50 values for the six rats between the PfCyRPA-I53-50 group and the PfCyRPA-EM-I53-50 group'

L255: Suggest rephrasing to 'When assessed in our pre-clinical model, IgG induced by PfCyRPA-EM-I53-50 immunisation were more specific to PfCyRPA but less abundant than those induced by PfCyRPA-I53-50, resulting in a similar parasite growth-inhibition potency.'

Throughout methods: Dynamic light scattering acronym (DLS) is specified multiple times in the methods (l346, 356 and 358). This only needs to be done once. Also, the text reads 'were confirmed dynamic light scattering' whereas should read were confirmed by dynamic light scattering'?

L284: space between the word and the reference.

L395: Define reservoir solution. Also, there is a mixture of 100 nL (capital on Litre) protein and reservoir solution. This sentence does not read well and I wonder whether a concentration is needed for the protein at the least, or the sentence needs re-writing.

L437: Please describe in more detail or provide a reference for this (Williams et al., 2024 perhaps?).

Methods: space between number and SI unit is inconsistent, particularly around the paragraphs starting Line 366, 375, 389, 405. Space should be added and all the methods checked.

General: the 50 in EC50 should be subscript.

Figure 2 title: too many and's.

Supplementary Figure 3: In the Figure, 'characterisation' is spelled incorrectly.

Responses to reviews:

Referee #2 (Comments on Novelty/Model System for Author):

This manuscript reports on the rational design and optimization of a blood-stage malaria vaccine targeting the *Plasmodium falciparum* cysteine-rich protective antigen (PfCyRPA), a key component of the PfPCRCR complex. The PCRCR complex is required for red blood cell invasion and includes Rh5 protein that is a target for a current blood stage vaccine Rh5.

To enhance immunogenicity, authors compared two nanoparticle display systems for presenting PfCyRPA: a post-production conjugation approach and a direct fusion strategy using the I53-50 nanoparticle scaffold. The direct fusion system proved superior, yielding higher titers of growth-inhibitory antibodies, improved manufacturability, and greater consistency.

For improved antibody quality, authors applied structure-guided immunogen design. They engineered a synthetic variant, PfCyRPA-EM, which focuses the immune response on the most protective epitopes (blades 1 and 2 of the β -propeller domain). By leveraging computational protein design (PROSS) and yeast display selection for growth-inhibitory antibody binding, they obtained a correctly folded, highly stable immunogen. PfCyRPA-EM demonstrated a 14°C increase in thermal stability, six-fold higher expression in mammalian cells, and compatibility with bacterial expression as an I53-50 fusion.

In a preclinical rat model, PfCyRPA-EM-I53-50 demonstrated improved quality as it was induced more efficacious parasite growth-inhibitory antibodies than the original PfCyRPA-I53-50. The enhanced stability and manufacturability of PfCyRPA-EM further highlighted its suitability for vaccine development and distribution.

Given the partial and non-durable protection of current leading candidates (e.g., RH5.1), the new PfCyRPA-EM immunogen represents a significant advance. It is now positioned for combination studies with other PfPCRCR components, with the aim of achieving more comprehensive and durable protection against malaria.

Key Innovations:

- Nanoparticle display optimization: Direct fusion to I53-50 for superior immunogenicity and manufacturability.
- Structure-based immunogen design: Focused epitope presentation for higher quality antibody responses.
- Enhanced manufacturability: Improved stability and expression in both mammalian and bacterial systems.
- Translational potential: PfCyRPA-EM is a strong candidate for next-generation malaria vaccines, particularly in multicomponent formulations.

This report exemplifies how advanced protein engineering and immunogen design can address both immunological and practical challenges in vaccine development.

We thank the reviewer for this accurate summary and positive comments about our project.

Referee #2 (Remarks for Author):

The manuscript is an excellent example of how advanced protein engineering and immunogen design can address both immunological and practical challenges in vaccine development. It is of high technical quality and employs cutting-edge approaches in protein design. My suggestions are minor and intended to further improve clarity and accessibility for nonspecialist readers.

1. Introduction: The authors provide a compelling rationale against using full-length proteins for immunogen design, noting that such proteins often contain epitopes that elicit non-functional antibodies, thereby diminishing the quality of the immune response. They also reference studies showing that the most inhibitory antibodies against PfCyRPA are directed against blades 1 and 2 of the β -propeller domain. However, it is surprising that the initial

Results section presents experiments using full-length PfCyRPA. I recommend revising the Introduction to include additional background on the nanoparticle platforms used, specifically describing the properties of HBsAg and I53-50. The differences between these two particles are not immediately apparent to nonspecialists, making it difficult to appreciate the methodological rationale. For example, how many immunogens are displayed per particle? Should this difference influence the choice of formulation dose?

As the primary advance in this study is the design of an improved PfCyRPA-based immunogen through structure-guided methods, we prefer not to change the introduction, but to keep the focus on rational immunogen design. However, we agree with the reviewer's point that this leads to a disjunction with the first results section. We have added a sentence at the start of the results in lines 90-91 to smooth this transition.

However, we have now included more information about the nature of the nanoparticles, specifying and providing a reference for the HBsAg particle (lines 105-108) and the highlighting the number of antigens on the I53-50 nanoparticle (line 113). In terms of dose, this should not matter as we calculated dose based on antigen concentration (lines 137-8).

Additionally, the rationale for removing potential PfCyRPA glycosylation sites is unclear and should be clarified.

We added a sentence to lines 101-103 to state that our use of mutations to remove glycosylation is due to the lack of N-linked glycans on Plasmodium surface proteins.

2. Discussion: This section largely reiterates the results rather than interpreting them. It would benefit from a more in-depth discussion of the key differences between the two nanoparticle platforms tested. For instance, is the observed inferiority of HBsAg attributable to conjugation efficiency or other intrinsic properties of the protein? What types of nanoparticles are currently available for vaccine development? Finally, it remains unclear how the efficacy of PfCyRPA-EM-I53-50 compares to the most potent version of Rh5. Is this new particle a significant advancement towards a highly effective blood-stage malaria vaccine, or is it better considered as a complementary component to Rh5?

We have considered our discussion and our preference is not to make substantial changes. We do provide analysis of the experiments, such as in lines 257-266, which discuss the pros and cons of the nanoparticle platforms used, but prefer not speculate further on reasons why one might be better.

In addition, as we have not run a direct comparison of PfCyRPA-EM vs any PfRH5-based immunogen in this study, we prefer not to speculate on their relative merits. We are conducting studies which combine these antigens and will report a properly controlled study in a future manuscript. We do already make the point that PfCyRPA-EM might be used in such combinations in lines 280-289.

Referee #3 (Comments on Novelty/Model System for Author):

Experimental reproducibility of some of the work (ELISA's, Growth assays) is a bit unclear. Since the conclusions are based on these assay readouts for one vaccine platform over another, it would be better to have stronger comparative data. The improved vaccine platform performance is a real positive. It would be good to explore whether two immunisations would be enough to generate a promising growth inhibitory immune response with the optimised vaccine. Medical impact will strengthen when used in combination with Rh5 antibodies, which are not done here.

We thank the reviewer for these perspectives and for highlighting that the improved performance is a real positive.

The reviewer highlights that two dose data as well as three dose data would be useful. We do present the ELISA data for responses after two and three doses (Figure 3C) showing no statistically significant difference in antigen-specific antibodies between these two responses.

One of the challenges, however, of the rat model, is the quantity of antibody required for the GIA assays, which can only be obtained at the experimental end point. To obtain two dose data would therefore require a new set of animals. In our view, particularly in view of the ELISA data, this is not justifiable in terms of animal welfare and that these dosing studies would be better done on transition to human clinical trials.

On the topic of combination with RH5-based vaccines, we fully agree, but these experiments are outside the scope of this study. We are currently testing PfCyRPA-based immunogens in combination with various PfRH5-based immunogens, designed using structure-guided methods to present epitopes for the most effective antibodies. We will look forward to sharing the outputs of these studies with the community when ready.

Referee #3 (Remarks for Author):

In this paper Alam et al. develop an immunogen against blood stage Plasmodium falciparum malaria based on PfCyRPA, part of a complex essential for erythrocyte invasion. The authors focus on blades 1 and 2 of PfCyRPA that previous studies have shown to be most immunogenic. Firstly, the paper compares two platforms for antigen presentation and show that direct fusion to I53-50 nanoparticles is simpler to produce and most immunogenic in an in-vitro parasite growth inhibition assay. The study proceeds to a structure guided design of blades 1 and 2 of PfCyRPA in silico to improve epitope folding and expression, however the resulting design forms a dimer and has a lower affinity to known PfCyRPA binding antibodies Cy.004 and Cy007. To improve this design, the authors then generate a yeast surface display library and select for binders of both Cy.004 and Cy.007 using MACS and several rounds of FACS. The resulting PfCyRPA-EM closely resembles the structure of PfCyRPA blades 1 and 2, but has a higher thermal stability than PfCyRPA. The optimised PfCyRPA-EM antigen is finally fused to I53-50 nanoparticles and tested in rats. Compared to PfCyRPA, PfCyRPA-EM induced a lower but more specific anti-PfCyRPA response, resulting in similar parasite growth inhibition.

Overall, this paper presents an improved design and more cost-effective production method of an antigen that structurally mimics blades 1 and 2 of PfCyRPA. The immunogenicity in rats is lower but more specific compared to PfCyRPA. The optimised antigen has the potential to have an additive effect in combination with other antigens to produce an invasion-blocking vaccine against P. falciparum malaria.

We thank the reviewer for this accurate assessment of our study.

Specific comments regarding yeast surface display:

- Figure 1 is titled 'Assembly, biophysical characterisation, and immunogenicity.....'. However, there really is not indication of the Assembly process, although it would be quite useful to run the reader through this. I would either change the title or, better yet, add in a diagram of the Assembly.

The reviewer is right as Figure 1 just shows immunogenicity data. We have therefore shortened the title. The data relating to assembly is in Figure S1 and we have added a diagram of the assembly into this figure as panel B.

In Figures 1, 3, 4, there is no indication of the number of repeat experiments (biological replicates) for assays such as growth assays and ELISAs. Although multiple samples are tested, it is not clear whether the data is from a single experiment or mean/median of multiple experiments. These are not difficult experiments and it is always best to do biological replicates in case the assays are impacted by problems in parasite growth, setup, readout, etc. Biological replicates strengthen the data and interpretability. Since there are no error bars, it seems likely that these experiments were done only once. Since there are outliers, it is not clear whether this represents less efficient antibody production after immunisation with a specific rat or some sort of problem with the assay for specific samples. It is reasonable for

replicates to be done for these experiments if they have not been and the combined data shown. Failing that, the number of repeats should be clearly stated so the reader can decide on the strength of the data.

For both ELISA and GIA data, each condition has n=6 biological replicates as the sera or IgG from each rat is treated independently. Therefore, each point or line represents data from a single animal. In addition, each point or line shows the average of three technical replicates. Showing each of these replicates for each of the points or lines would make the figures very crowded and so we do not represent them in this way. However, each of these triplicate measurements for each data point is included in the source data, allowing transparent sharing of the raw data should readers wish to assess data quality and consistency. We have added some sentences to the legends of Figures 1 and 3 to more clearly make these points.

Please show the gating strategy in Supplementary Figure 3. As presented it appears that the total population from each round is fed into the next round, which contrasts with the figure legend that suggests less than 1% of cells were selected to the next round.

As suggested, we edited new supplementary figure 3, to include the gates used for each round as red rectangle and we have labelled each with the % of cells which were selected to proceed to the next round of sorting.

In the methods it is unclear what happens in between each sorting round. Please add detail.

As suggested, we have added a new section into the methods, in lines 353-362 to explain in more detail the nature of each round of sorting and the changes that were made between rounds.

Was there a rationale for starting the selection with Cy.004 rather than Cy.007?

We used Cy.004 for early rounds of gating as it has the highest growth-inhibitory activity of the CyRPA monoclonal antibodies. Cy.007 was used for subsequent rounds to ensure selection of immunogens which recapitulate both surfaces. We have added a sentence to explain this in lines 175-78.

Figure 2C does not show convincing selection of higher affinity binding. In fact, there is a population with high expression/low binding that appears to be enriched. As shown in Supplementary Figure 3 this non-binding population is lost during Cy.004 selecting rounds but re-appears in round 6 after switching to selection with Cy.007. Perhaps you could graph the percent of cells in the high expression/high binding gate in the non-mutagenized library, the mutagenized library pre-selection and the mutagenized library post selection? A more convincing illustration of the enrichment would be a contrast between the mutagenized library pre and post selection.

We have adjusted Figure 2C to clarify this. We have added a black dotted rectangle in the top right corner, which represents the high expression and high binding clones which we were aiming to select. We also include the number of cells in this box for the starting point library (PROSS design library) and the final library (Selected mutagenesis library). This shows that selection from 0.02% or 0.03% of high expression, high binding clones in the starting library to 4.93% or 5.83% for Cy.004 and Cy.007 respectively. This shows clear selection of the high expression, high-binding clones which we were aiming to select.

Can the authors discuss the implications for a larger fraction of growth inhibitory antibodies targeting CyRPA for the PfCyRPA-EM-153-50 immunised samples, even as higher non-inhibitory antibody titres overall mean that the growth inhibitory activity is lower when compared as a proportion of total IgG? Is there a way that these could be selected for to improve CyRPA inhibitory antibody concentration through immunisation? What happens if a bleed post second immunisation is tested, do we see equivalent or proportionally improved CyRPA growth inhibitory antibody levels relative to total IgG or CyRPA antibody activity,

even though it does not reach the peak seen after the 3rd immunisation? It would be good to do this. If it was the case that the 2nd immunisation provides the best CyRPA protective response, could immunisation be optimised around 2 vaccinations in preference to 3 where more non-specific/inhibitory IgG seems to increase, but not CyRPA growth inhibitory antibodies.

We have addressed this point earlier. In brief, we do present the ELISA data for responses after two and three doses (Figure 3C) showing no statistically significant change in PfCyRPA-reactive antibody levels between a second and a third dose, against either PfCyRPA or PfCyRPA-EM. However, to obtain two dose GIA data would require immunisation of a new cohort of animals. In our view, taking into account the ELISA data, this is not justifiable in terms of animal welfare and that these types of dose studies would be better done on transition to human clinical trials.

The immunogenicity of PfCyRPA-EM is qualitatively similar to that reported by Bjornsson et al. (2024) i.e. immunofocussing of neutralizing antibodies at the expense of overall anti-PfCyRPA antibody titres. The authors mention the limitations of the design of the Bjornsson et al. (2024) study in the introduction, however a comparison of the results between these two studies in the discussion would clarify the impact of the findings of this paper.

We do not wish to be over-critical of Bjornsson et al, or to suggest that our study is a response to theirs and so we have kept cross-comparison to a minimum. However, we have now added a single sentence in lines 230-234 to provide a comparison. The comparable data is our reengineered PfCyRPA blades 1 and 2 construct and their Fragment A, which consists of the same region of PfCyRPA without reengineering. In their study, they show that antibodies induced by Fragment A have an EC50 for growth inhibition at least 10-fold weaker than that for antibodies induced by PfCyRPA. In our study, there is no statistical difference in EC50 for antibodies induced by PfCyRPA-EM and PfCyRPA, highlighting the value in re-engineering immunogens to correctly display the epitope surface.

Minor specific comments:

L169 - This paragraph has very frequent references to supplementary figures and compares data between the main Figure 2 and supplementary figures. Suggest incorporating some of the supplementary data in Figure 2.

We agreed with the reviewer that it would be useful for the reader to have the SPR data for CyRPA and CyRPA-EM in the same figure to allow direct comparison and so have moved the CyRPA data into Figure 2. The data for intermediates in the design process remain in supplementary data.

L172 and L173 - Supplementary Figure 2C and Figure 2C labels refer to PfCyRPA-EM, not BDP-YD1-4, even though new nomenclature hasn't been introduced yet. Supplementary Table 2 refers to Yeast display design 3 / BDP-YD1-4 / PfCyRPA-EM. It would be clearer to give the antigen one name throughout.

On reflection, we agree with the reviewer that this is a little confusing. To us, it made sense as we made a set of BDP versions and then selected PfCyRPA-EM based on the data, but we have nevertheless taken the reviewers advice and changed this nomenclature.

L190: Missing word I53-50A.1NT1 was expressed

We have added 'was'.

L191: Is the cost effectiveness due to higher expression in Expi293F cells or due to the ability to express in E. coli? As phrased, this is unclear.

We have clarified that both findings indicate a potentially benefit for cost-effectiveness.

L175 : Although much lower than for the PROSS design, the KD of PfCyRPA-EM vs Cy.007 is still approx. 3x higher than the KD for PfCyRPA, meaning there is lower affinity. Do you think this suggests some misfolding?

We have added a little more data and explanation to the manuscript to address this question. In particular, we have expanded this section of the text (lines 186-189) to more explicitly state that, while the Cy.004 affinities are very similar, the affinity for Cy.007 is three-fold different due to a higher off-rate. We also included in Figure 2F a new close-up view of the alignment of CyRPA and CyRPA-EM in both Cy.004 and Cy.007 epitopes, showing no significant difference in structure. The difference in affinity for Cy.007 is therefore not due to misfolding. However, it could be due to increased flexibility of this region of CyRPA-EM when out of the context of CyRPA, leading to increased off-rates of binding. We speculate along these lines in lines 195-198.

L192: Missing word: I53-50A.1NT1 was assembled

We have added 'was'.

Figure 4D: Cy.003 binding is missing in bottom panel.

The reviewer is correct. This is because the Cy.003 epitope is not fully recapitulated in blade 1 and 2, although most the epitope surface is present. We therefore used Cy.004 and Cy.007 throughout.

L213: Suggest rephrase to 'there was no statistical difference between the mean EC50 values for the six rats between the PfCyRPA-I53-50 group and the PfCyRPA-EM-I53-50 group' .

We thank the reviewer for this sensible suggestion, which we have followed.

L255: Suggest rephrasing to 'When assessed in our pre-clinical model, IgG induced by PfCyRPA-EM-I53-50 immunisation were more specific to PfCyRPA but less abundant than those induced by PfCyRPA-I53-50, resulting in a similar parasite growth-inhibition potency.'

We thank the reviewer for this sensible suggestion, which we have followed.

Throughout methods: Dynamic light scattering acronym (DLS) is specified multiple times in the methods (I346, 356 and 358). This only needs to be done once. Also, the text reads 'were confirmed dynamic light scattering' whereas should read 'were confirmed by dynamic light scattering'?

We have removed this repetition and now DLS is defined at first use in the text only (line 119).

L284: space between the word and the reference.

We have fixed this.

L395: Define reservoir solution. Also, there is a mixture of 100 nL (capital on Litre) protein and reservoir solution. This sentence does not read well and I wonder whether a concentration is needed for the protein at the least, or the sentence needs re-writing.

We accidentally used 'well solution' and 'reservoir solution' for the same solution. This has now been fixed and the sentence has been clarified in other ways.

L437: Please describe in more detail or provide a reference for this (Williams et al., 2024 perhaps?).

We have added Williams et al, as suggested.

Methods: space between number and SI unit is inconsistent, particularly around the paragraphs starting Line 366, 375, 389, 405. Space should be added and all the methods checked.

We are confident that these will be put into house style during type-setting.

General: the 50 in EC50 should be subscript.

We have corrected these.

Figure 2 title: too many and's.

We have removed the errant first 'and'.

Supplementary Figure 3: In the Figure, 'characterisation' is spelled incorrectly.

This typo has now been corrected.

17th Dec 2025

Dear Dr. Higgins,

Thank you for submitting your revised study. We have now received the reports from referees #2 and #3. As you will see below, they are satisfied with the revisions, and I will therefore be able to accept your manuscript once the following editorial concerns are addressed:

1/ Manuscript text:

- Please indicate in track changes mode any new modification.
- We think the format of your article is better suited to a Report, and reassigned the article category accordingly. There is no additional action needed on your side.
- Please correct the order and headings of the manuscript sections to: Abstract / Keywords / The Paper Explained / Introduction / Results / Discussion / Methods / Data Availability / Acknowledgements / Disclosure and Competing Interests Statement / References / Main Figure Legends / Expanded View Figure Legends.
- Please provide up to 5 keywords.
- Please provide the following information in the Methods, and make sure to fill the author checklist accordingly:
 - o Cells: please indicate the origin of the cells, and whether the cells were tested for mycoplasma contamination.
 - o Animals: please indicate the origin as well as the housing and husbandry conditions. Please state details of authority granting ethics approval (IRB or equivalent committee(s), provide reference number for approval. Include a statement of compliance with ethical regulations.
 - o Human samples: if applicable, please state details of authority granting ethics approval, include statements on informed consent and Helsinki declaration.
 - o Antibodies: please provide dilutions/concentrations.
 - o Statistics: please provide a statement on randomization and inclusion/exclusion criteria.
- Data availability section: please note that the data must be publicly accessible before acceptance of the manuscript and a URL provided. Please remove "and the remaining data is provided as source data".
- Acknowledgements: the fundings listed in the manuscript should match the information provided in the submission system, please adjust accordingly (currently, not provided in the manuscript text).
- Please rename "Competing interests" to "Disclosure and competing interests statement".

2/ Figures:

- Please remove the EV tables from the manuscript text file and upload them as separate files, one per table.
- Please check the labeling in the figures, as some titles to be inserted twice on top of each other.
- Please note that information related to n is missing in the legends of figures 3B, C, E, G.

3/ Source Data:

Please upload the files as one zip folder per figure - inside each zip folder, the data should be organized into subfolders, with one subfolder for each figure panel.

4/ As part of the EMBO Publications transparent editorial process initiative (see our Editorial at <http://embomolmed.embopress.org/content/2/9/329>), EMBO Molecular Medicine will publish online a Review Process File (RPF) to accompany accepted manuscripts.

This file will be published in conjunction with your paper and will include the anonymous referee reports, your point-by-point response and all pertinent correspondence relating to the manuscript. Let us know whether you agree with the publication of the RPF.

Looking forward to the submission of your revised files,

With kind regards,

Lise Roth

To submit your manuscript, please follow this link:

***** Reviewer's comments *****

Referee #2 (Comments on Novelty/Model System for Author):

This manuscript describes the design and tuning of a blood-stage malaria vaccine aimed at PfCyRPA, a key part of the PfPCRCR complex. That complex is required for red blood cell invasion. It includes Rh5, the target of a current blood-stage vaccine effort.

To boost immunogenicity, the authors tested two nanoparticle display strategies for PfCyRPA: post-production conjugation and direct fusion using the I53-50 scaffold. Direct fusion came out on top. It gave higher titers of growth-inhibitory antibodies, was easier to manufacture, and was more consistent.

They also worked on antibody quality. Using structure-guided design, they focused the immune response on the most protective sites-blades 1 and 2 of the β -propeller domain. The result is a synthetic variant, PfCyRPA-EM. It was built with computational design (PROSS) and yeast display selections for binding to growth-inhibitory antibodies. The protein folds correctly and is highly stable. PfCyRPA-EM showed better thermal stability, higher expression in mammalian cells, and for I53-50 fusion, compatibility with bacteria expression system.

In immunization experiments in rats, the current preclinical model system, PfCyRPA-EM-I53-50 induced antibodies that were more effective at inhibiting parasite growth than the original PfCyRPA-I53-50. Its improved stability and manufacturability also point to smoother development and distribution.

Given the partial, short-lived protection from today's leading candidates (like RH5.1), PfCyRPA-EM looks like a real step forward. It's now ready for combination studies with other PfPCRCR components to push toward broader, more durable protection against malaria.

Key innovations:

1. Nanoparticle display optimization: direct fusion to I53-50 improved immunogenicity and manufacturability.
2. Structure-based immunogen design: focused epitope presentation yielded higher-quality antibody responses.
3. Enhanced manufacturability: better stability and higher expression in mammalian cells; compatible with bacterial expression.
4. Translational potential: a strong candidate for next-generation malaria vaccines, possibility to use in multicomponent formulations.

Overall, this work shows how modern protein engineering and immunogen design can tackle both immune and practical hurdles in vaccine development.

Referee #2 (Remarks for Author):

Most of my comments were addressed by the authors and I have no additional remarks.

Referee #3 (Comments on Novelty/Model System for Author):

Experimental reproducibility has improved and we recognise the limitations of gaining more samples through animal vaccination. The improved vaccine platform performance is a real positive. It would be good to explore whether two immunisations would be enough to generate a promising growth inhibitory immune response with the optimised vaccine. Medical impact will strengthen when used in combination with Rh5 antibodies, which are not done here.

The authors addressed the remaining editorial issues.

14th Jan 2026

Dear Dr. Higgins,

Thank you for submitting your revised files. I am pleased to inform you that your manuscript is accepted for publication and is now being sent to our publisher to be included in the next available issue of EMBO Molecular Medicine.

You may qualify for financial assistance for your publication charges - either via a Springer Nature fully open access agreement or an EMBO initiative. Check your eligibility: <https://link.springer.com/journal/44321/how-to-publish-with-us>

With my best wishes for 2026,

Lise Roth

>>> Please note that it is EMBO Molecular Medicine policy for the transcript of the editorial process (containing referee reports and your response letter) to be published as an online supplement to each paper. If you do NOT want this, you will need to inform the Editorial Office via email immediately. More information is available here: <https://link.springer.com/partners/embo-press/editorial-policies#Peer%20review>